# Breaking antimicrobial resistance by disrupting extracytoplasmic protein folding

R Christopher D Furniss[1†], Nikol Kaderabkova[1,2†], Declan Barker[1], Patricia Bernal[3], Evgenia Maslova[4], Amanda AA Antwi[1], Helen E McNeil[5], Hannah L Pugh[5], Laurent Dortet[1,6,7,8], Jessica MA Blair[5], Gerald Larrouy-Maumus[1], Ronan R McCarthy[4], Diego Gonzalez[9], Despoina AI Mavridou[1,2,10]*

[1]MRC Centre for Molecular Bacteriology and Infection, Department of Life Sciences, Imperial College London, London, United Kingdom; [2]Department of Molecular Biosciences, University of Texas at Austin, Austin, United States; [3]Department of Microbiology, Faculty of Biology, Universidad de Sevilla, Seville, Spain; [4]Division of Biosciences, Department of Life Sciences, College of Health and Life Sciences, Brunel University London, Uxbridge, United Kingdom; [5]Institute of Microbiology and Infection, College of Medical and Dental Sciences, University of Birmingham, Birmingham, United Kingdom; [6]Department of Bacteriology-Hygiene, Bicêtre Hospital, Assistance Publique - Hôpitaux de Paris, Le Kremlin-Bicêtre, France; [7]EA7361 "Structure, Dynamics, Function and Expression of Broad-spectrum β-lactamases", Paris-Sud University, LabEx Lermit, Faculty of Medicine, Le Kremlin-Bicêtre, France; [8]French National Reference Centre for Antibiotic Resistance, Le Kremlin-Bicêtre, France; [9]Laboratoire de Microbiologie, Institut de Biologie, Université de Neuchâtel, Neuchâtel, Switzerland; [10]John Ring LaMontagne Center for Infectious Diseases, University of Texas at Austin, Austin, United States

*For correspondence:
despoina.mavridou@austin.
utexas.edu

†These authors contributed
equally to this work

Competing interest: The authors
declare that no competing
interests exist.

Reviewing Editor: Melanie
Blokesch, Ecole Polytechnique
Fédérale de Lausanne,
Switzerland

**Abstract** Antimicrobial resistance in Gram-negative bacteria is one of the greatest threats to global health. New antibacterial strategies are urgently needed, and the development of antibiotic adjuvants that either neutralize resistance proteins or compromise the integrity of the cell envelope is of ever-growing interest. Most available adjuvants are only effective against specific resistance proteins. Here, we demonstrate that disruption of cell envelope protein homeostasis simultaneously compromises several classes of resistance determinants. In particular, we find that impairing DsbA-mediated disulfide bond formation incapacitates diverse β-lactamases and destabilizes mobile colistin resistance enzymes. Furthermore, we show that chemical inhibition of DsbA sensitizes multidrug-resistant clinical isolates to existing antibiotics and that the absence of DsbA, in combination with antibiotic treatment, substantially increases the survival of *Galleria mellonella* larvae infected with multidrug-resistant *Pseudomonas aeruginosa*. This work lays the foundation for the development of novel antibiotic adjuvants that function as broad-acting resistance breakers.

## Editor's evaluation

This work is based on the idea that targeting protein stability or inhibiting proper protein folding in the membrane/periplasmic space might abolish antimicrobial resistances (AMR) in Gram-negative bacteria. By targeting the primary disulfide bond formation enzyme DsbA, the authors provide a proof-of-principle for the inhibition of β-lactamases, MCR enzymes, and RND efflux pumps of model bacterial species as well as clinical isolates. Collectively, the study shows that a chemical inhibitor of

the DSB system sensitizes resistant bacteria to several antibiotics such as β-lactams, chloramphenicol, and colistin.

## Introduction

Antimicrobial resistance (AMR) is one of the most important public health concerns of our time (*Rochford et al., 2018*). With few new antibiotics in the pharmaceutical pipeline and multidrug-resistant bacterial strains continuously emerging, it is more important than ever to develop novel antibacterial strategies and find alternative ways to break resistance. While the development of new treatments for Gram-negative bacteria is considered critical by the WHO (*Tacconelli et al., 2018*), identifying novel approaches to target these organisms is particularly challenging due to their unique double-membrane permeability barrier and the vast range of AMR determinants they produce. For this reason, rather than targeting cytoplasmic processes, antimicrobial strategies that inhibit cell-envelope components or impair the activity of resistance determinants are being increasingly pursued (*Hart et al., 2019*; *Laws et al., 2019*; *Luther et al., 2019*; *Nicolas et al., 2019*; *Srinivas et al., 2010*).

The Gram-negative cell envelope is home to many different AMR determinants, with β-lactamase enzymes currently posing a seemingly insurmountable problem. More than 6500 unique enzymes capable of degrading β-lactam compounds have been identified to date (*Supplementary file 1*). Despite the development of more advanced β-lactam antibiotics, for example the carbapenems and monobactams, resistance has continued to emerge through the evolution of many broad-acting β-lactamases (*Bush, 2018*). This constant emergence of resistance not only threatens β-lactams, the most commonly prescribed antibiotics worldwide (*Meletis, 2016*; *Versporten et al., 2018*), but also increases the use of last-resort agents, like the polymyxin antibiotic colistin, for the treatment of multidrug-resistant infections (*Li et al., 2006*). As a result, resistance to colistin is on the rise, due in part to the alarming spread of novel cell-envelope colistin resistance determinants. These proteins, called mobile colistin resistance (MCR) enzymes, represent the only mobilizable mechanism of polymyxin resistance reported to date (*Poirel et al., 2017*). Since their discovery in 2015, 10 families of MCR proteins have been identified and these enzymes are quickly becoming a major threat to the longevity of colistin (*Sun et al., 2018*). Alongside β-lactamases and MCR enzymes, Resistance-Nodulation-Division (RND) efflux pumps further enrich the repertoire of AMR determinants in the cell envelope. These multi-protein assemblies span the periplasm and remove many antibiotics (*Blair et al., 2014*; *Cox and Wright, 2013*), rendering Gram-negative bacteria inherently resistant to important antimicrobials.

Inhibition of AMR determinants has traditionally been achieved through the development of antibiotic adjuvants. These molecules impair the function of resistance proteins and are used in combination with existing antibiotics to eliminate challenging infections (*Laws et al., 2019*). Whilst this approach has proven successful and has led to the deployment of several β-lactamase inhibitors that are used clinically (*Laws et al., 2019*), it has so far not been able to simultaneously incapacitate different classes of AMR determinants. This is because traditional antibiotic adjuvants bind to the active site of a resistance enzyme and thus are only effective against specific protein families. To disrupt AMR more broadly, new strategies have to be developed that target the biogenesis or stability, rather than the activity, of resistance determinants. In this way, the formation of multiple resistance proteins can be inhibited at once, instead of developing specific compounds that inactivate individual AMR enzymes after they are already in place.

In extracytoplasmic environments protein stability often relies on the formation of disulfide bonds between cysteine residues (*Goemans et al., 2014*; *Heras et al., 2007*). Notably, in the cell envelope of Gram-negative bacteria this process is performed by a single pathway, the DSB system, and more specifically by a single protein, the thiol oxidase DsbA (*Bardwell et al., 1991*; *Denoncin and Collet, 2013*; *Hiniker and Bardwell, 2004*; *Kadokura et al., 2004*; *Martin et al., 1993*). DsbA has been shown to assist the folding of hundreds of proteins in the periplasm (*Kadokura et al., 2004*; *Dutton et al., 2008*; *Vertommen et al., 2008*; *Figure 1A*), including a vast range of virulence factors (*Heras et al., 2009*; *Landeta et al., 2018*). As such, inhibition of DSB proteins has been proposed as a promising broad-acting strategy to target bacterial pathogenesis without impairing bacterial viability (*Denoncin and Collet, 2013*; *Heras et al., 2009*; *Landeta et al., 2018*; *Heras et al., 2015*). Nonetheless, the contribution of oxidative protein folding to AMR has never been examined. Since several cell

**eLife digest** Antibiotics, like penicillin, are the foundation of modern medicine, but bacteria are evolving to resist their effects. Some of the most harmful pathogens belong to a group called the 'Gram-negative bacteria', which have an outer layer – called the cell envelope – that acts as a drug barrier. This envelope contains antibiotic resistance proteins that can deactivate or repel antibiotics or even pump them out of the cell once they get in. One way to tackle antibiotic resistance could be to stop these proteins from working.

Proteins are long chains of building blocks called amino acids that fold into specific shapes. In order for a protein to perform its role correctly, it must fold in the right way. In bacteria, a protein called DsbA helps other proteins fold correctly by holding them in place and inserting links called disulfide bonds. It was unclear whether DsbA plays a role in the folding of antibiotic resistance proteins, but if it did, it might open up new ways to treat antibiotic resistant infections.

To find out more, Furniss, Kaderabkova et al. collected the genes that code for several antibiotic resistance proteins and put them into *Escherichia coli* bacteria, which made the bacteria resistant to antibiotics. Furniss, Kaderabkova et al. then stopped the modified *E. coli* from making DsbA, which led to the antibiotic resistance proteins becoming unstable and breaking down because they could not fold correctly.

Further experiments showed that blocking DsbA with a chemical inhibitor in other pathogenic species of Gram-negative bacteria made these bacteria more sensitive to antibiotics that they would normally resist. To demonstrate that using this approach could work to stop infections by these bacteria, Furniss, Kaderabkova et al. used Gram-negative bacteria that produced antibiotic resistance proteins but could not make DsbA to infect insect larvae. The larvae were then treated with antibiotics, which increased their survival rate, indicating that blocking DsbA may be a good approach to tackling antibiotic resistant bacteria.

According to the World Health Organization, developing new treatments against Gram-negative bacteria is of critical importance, but the discovery of new drugs has ground to a halt. One way around this is to develop ways to make existing drugs work better. Making drugs that block DsbA could offer a way to treat resistant infections using existing antibiotics in the future.

envelope AMR determinants contain multiple cysteines (*Bardwell et al., 1991*; *Piek et al., 2014*), we hypothesized that interfering with the function of DsbA would not only compromise bacterial virulence (*Heras et al., 2015*), but might also offer a broad approach to break resistance across different mechanisms by affecting the stability of resistance proteins. Here, we test this hypothesis by investigating the contribution of disulfide bond formation to three of the most important resistance mechanisms in the cell envelope of Enterobacteria: the breakdown of β-lactam antibiotics by β-lactamases, polymyxin resistance arising from the production of MCR enzymes and intrinsic resistance to multiple antibiotic classes due to RND efflux pumps. We find that some of these resistance mechanisms depend on DsbA and we demonstrate that when DsbA activity is chemically inhibited, resistance can be abrogated for several clinically important enzymes. Our findings prove that targeting protein homeostasis in the cell envelope allows the impairment of diverse AMR proteins and therefore could be a promising avenue for the development of next-generation therapeutic approaches.

## Results

### The activity of multiple cell envelope resistance proteins is dependent on DsbA

DsbA has been shown to assist the folding of numerous periplasmic and surface-exposed proteins in Gram-negative bacteria (*Figure 1A*; *Heras et al., 2009*; *Landeta et al., 2018*; *Heras et al., 2015*). As many AMR determinants also transit through the periplasm, we postulated that inactivation of the DSB system may affect their folding, and therefore impair their function. To test this, we first focused on resistance proteins that are present in the cell envelope and contain two or more cysteine residues, since they may depend on the formation of disulfide bonds for their stability and folding (*Bardwell et al., 1991*; *Piek et al., 2014*). We selected a panel of 12 clinically important β-lactamases

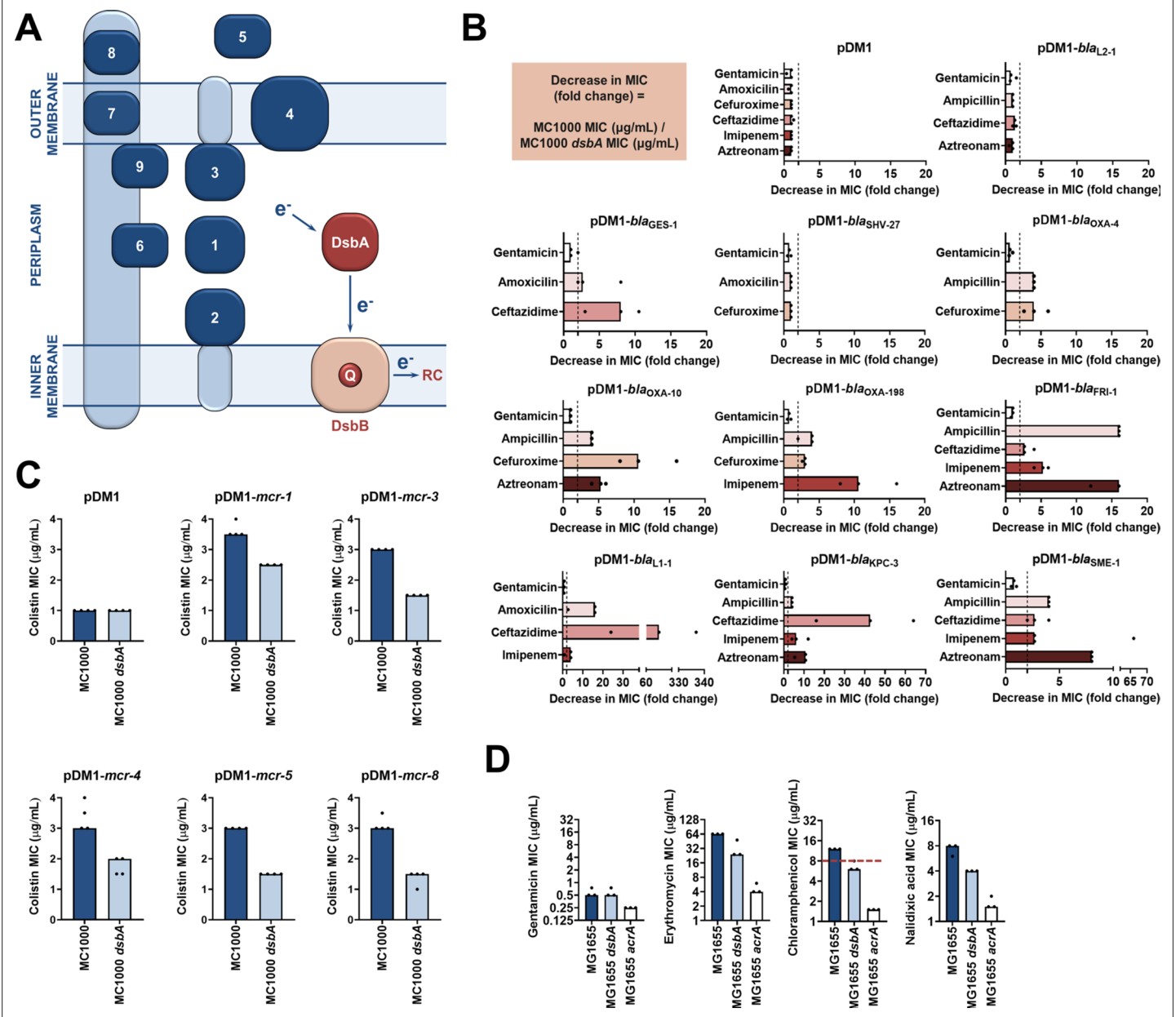

**Figure 1.** Several antimicrobial resistance mechanisms depend on disulfide bond formation. (**A**) DsbA introduces disulfide bonds into extracytoplasmic proteins containing two or more cysteine residues. After each round of oxidative protein folding, DsbA is regenerated by the quinone (Q)-containing protein DsbB, which in turn transfers the reducing equivalents to the respiratory chain (RC) (*Kadokura et al., 2003*). DsbA substrates (in dark blue) are distributed throughout the extracytoplasmic space of Gram-negative bacteria. Disulfides are introduced to (1) soluble periplasmic proteins (e.g. alkaline phosphatase, β-lactamases; *Bardwell et al., 1991*), (2) periplasmic domains of inner-membrane proteins (e.g LptA-like enzymes (*Piek et al., 2014*), (3) periplasmic domains of outer-membrane proteins (e.g. RcsF; *Denoncin and Collet, 2013*), (4) outer-membrane proteins (e.g. OmpA, LptD; *Denoncin and Collet, 2013*; *Heras et al., 2009*), (5) secreted proteins (e.g. toxins or enzymes; *Heras et al., 2009*), (6–9) protein components of macromolecular assemblies like secretion systems, pili or flagella (*Heras et al., 2009*) (e.g. (6) GspD, (7) EscC, (8) BfpA, (9) FlgI); all examples are *E. coli* proteins with the exception of LptA. (**B**) β-lactam MIC values for *E. coli* MC1000 expressing diverse disulfide-bond-containing β-lactamases (Ambler classes A, B and D) are substantially reduced in the absence of DsbA (MIC fold changes: > 2, fold change of 2 is indicated by the black dotted lines); no effect is observed for SHV-27, which is further discussed in *Figure 1—figure supplement 3*. DsbA dependence is conserved within phylogenetic groups (see *Figure 1—figure supplement 2*). No changes in MIC values are observed for the aminoglycoside antibiotic gentamicin (white bars) confirming that absence of DsbA does not compromise the general ability of this strain to resist antibiotic stress. No changes in MIC values are observed for strains harboring the empty vector control (pDM1) or those expressing the class A β-lactamase L2-1, which contains three cysteines but no disulfide bond (top row). Graphs show MIC fold changes for β-lactamase-expressing *E. coli* MC1000 and its *dsbA* mutant from three biological experiments each conducted as a single technical repeat; the MIC values used to generate this panel are presented in *Supplementary file 2a*. (**C**) Colistin MIC values for

*Figure 1 continued on next page*

*Figure 1 continued*

*E. coli* MC1000 expressing diverse MCR enzymes (*Figure 1—figure supplement 1*) are reduced in the absence of DsbA. Graphs show MIC values (µg/mL) from four biological experiments, each conducted in technical quadruplicate, to demonstrate the robustness of the observed effects. Gentamicin control data are presented in *Figure 1—figure supplement 6*. (**D**) Deletion of *dsbA* reduces the erythromycin, chloramphenicol and nalidixic acid MIC values for *E. coli* MG1655, but no effects are detected for the non-substrate antibiotic gentamicin. The essential pump component AcrA serves as a positive control. Graphs show MIC values (µg/mL) from three biological experiments, each conducted as a single technical repeat. Red dotted lines indicate the EUCAST clinical breakpoint for chloramphenicol.

The online version of this article includes the following figure supplement(s) for figure 1:

**Figure supplement 1.** Phylogenetic analysis of MCR- and EptA-like enzymes found in *Proteobacteria*.

**Figure supplement 2.** DsbA dependence is conserved within phylogenetic groups of disulfide-bond-containing β-lactamases.

**Figure supplement 3.** SHV-27 function is dependent on DsbA at temperatures higher than 37°C.

**Figure supplement 4.** Complementation of *dsbA* restores the β-lactam MIC values for *E. coli* MC1000 *dsbA* expressing β-lactamases.

**Figure supplement 5.** Complementation of *dsbA* restores the colistin MIC values for *E. coli* MC1000 *dsbA* expressing MCR enzymes.

**Figure supplement 6.** Gentamicin MIC values for *E. coli* MC1000 strains expressing MCR enzymes.

**Figure supplement 7.** Deletion of *dsbA* has no effect on membrane permeability in *E. coli* MC1000.

**Figure supplement 8.** Complementation of *dsbA* restores efflux-pump substrate MIC values for *E. coli* MG1655 *dsbA*.

**Figure supplement 9.** Deletion of *dsbA* has no effect on membrane permeability in *E. coli* MG1655.

**Table 1.** Overview of the β-lactamase enzymes investigated in this study.

Enzymes GES-1, –2 and –11 as well as KPC-2 and –3 belong to the same phylogenetic cluster (GES-42 and KPC-44, respectively, see *Supplementary file 1*). All other tested enzymes belong to distinct phylogenetic clusters (*Supplementary file 1*). The 'Cysteine positions' column states the positions of cysteine residues after position 30 and hence, does not include amino acids that would be part of the periplasmic signal sequence. All β-lactamase enzymes except L2-1 (shaded in grey; PDB ID: 1O7E) have one disulfide bond. The 'Mobile' column refers to the genetic location of the β-lactamase gene; 'yes' indicates that the gene of interest is located on a plasmid, while 'no' refers to chromosomally encoded enzymes. All tested enzymes have a broad hydrolytic spectrum and are either Extended Spectrum β-Lactamases (ESBLs) or carbapenemases. The 'Inhibition' column refers to classical inhibitor susceptibility that is, susceptibility to inhibition by clavulanic acid, tazobactam, or sulbactam.

| Enzyme | Amblerclass | Cysteine positions | Mobile | Spectrum | Inhibition |
|--------|-------------|--------------------|--------|----------|------------|
| L2-1 | A | C82 C136 C233 | no | ESBL | yes |
| GES-1 | A | C63 C233 | yes | ESBL | yes |
| GES-2 | A | C63 C233 | yes | ESBL | yes |
| GES-11 | A | C63 C233 | yes | Carbapenemase | yes |
| SHV-27 | A | C73 C119 | no | ESBL | yes |
| OXA-4 | D | C43 C63 | yes | ESBL | yes |
| OXA-10 | D | C44 C51 | yes | ESBL | no (*Aubert et al., 2001*) |
| OXA-198 | D | C116 C119 | yes | Carbapenemase | no (*El Garch et al., 2011*) |
| FRI-1 | A | C68 C238 | yes | Carbapenemase | no (*Dortet et al., 2015*) |
| L1-1 | B3 | C239 C267 | no | Carbapenemase | no (*Palzkill, 2013*) |
| KPC-2 | A | C68 C237 | yes | Carbapenemase | no (*Papp-Wallace et al., 2010*) |
| KPC-3 | A | C68 C237 | yes | Carbapenemase | no (*Papp-Wallace et al., 2010*) |
| SME-1 | A | C72 C242 | no | Carbapenemase | yes |

from different Ambler classes (classes A, B, and D), most of which are encoded on plasmids (*Table 1*). The chosen enzymes represent different protein structures, belong to discrete phylogenetic families (*Supplementary file 1*) and have distinct hydrolytic activities ranging from the degradation of penicillins and first, second and third generation cephalosporins (extended spectrum β-lactamases, ESBLs) to the inactivation of last-resort β-lactams (carbapenemases). In addition to β-lactamases, we selected five representative phosphoethanolamine transferases from throughout the MCR phylogeny (*Figure 1—figure supplement 1*) to gain a comprehensive overview of the contribution of DsbA to the activity of these colistin-resistance determinants.

We expressed our panel of 17 discrete resistance enzymes in an *Escherichia coli* K-12 strain (*E. coli* MC1000) and its isogenic *dsbA* mutant (*E. coli* MC1000 *dsbA*) and recorded minimum inhibitory concentration (MIC) values for β-lactam or polymyxin antibiotics, as appropriate. We found that the absence of DsbA resulted in a substantial decrease in MIC values ( > 2 fold cutoff) for all but one of the tested β-lactamases (*Figure 1B*, *Figure 1—figure supplement 2*, *Supplementary file 2a*). For the β-lactamase that seemed unaffected by the absence of DsbA, SHV-27, we performed the same experiment under temperature stress conditions (at 43 °C rather than 37 °C). Under these conditions, the lack of DsbA also resulted in a noticeable drop in the cefuroxime MIC value (*Figure 1—figure supplement 3*). A similar effect has been described for TEM-1, whereby its disulfide bond becomes important for enzyme function under stress conditions (temperature or pH stress) (*Schultz et al., 1987*). As SHV-27 has the narrowest hydrolytic spectrum out of all the enzymes tested, this result suggests that there could be a correlation between the hydrolytic spectrum of the β-lactamase and its dependence on DsbA for conferring resistance. In the case of colistin MICs, we did not implement a > 2 fold cutoff for observed decreases in MIC values as we did for strains expressing β-lactamases. Polymyxin antibiotics have a very narrow therapeutic window, and there is significant overlap between therapeutic and toxic plasma concentrations of colistin (*Nation et al., 2016*; *Plachouras et al., 2009*). Since patients that depend on colistin treatment are often severely ill, have multiple co-morbidities and are at high risk of acute kidney injury due to the toxicity of colistin, any reduction in the dose of colistin needed to achieve therapeutic activity is considered to be of value (*Nation et al., 2019*). Expression of MCR enzymes in our wild-type *E. coli* K-12 strain resulted in colistin resistance (MIC of 3 μg/mL or higher), while the strain harboring the empty vector was sensitive to colistin (MIC of 1 μg/mL). In almost all tested cases, the absence of DsbA caused re-sensitization of the strains, as defined by the EUCAST breakpoint (*E. coli* strains with an MIC of 2 μg/mL or below are classified as susceptible; *Figure 1C*), indicating that DsbA is important for MCR function. Taking into consideration the challenges when using colistin therapeutically (*Nation et al., 2016*; *Plachouras et al., 2009*; *Nation et al., 2019*), we conclude that deletion of *dsbA* leads to clinically meaningful decreases in colistin MIC values for the tested MCR enzymes (*Figure 1C*) and that the role of DsbA in MCR function should be further investigated.

Wild-type MIC values could be restored for all tested cysteine-containing enzymes by complementation of *dsbA* (*Figure 1—figure supplements 4 and 5*). Moreover, since DsbA acts on its substrates post-translationally, we performed a series of control experiments designed to assess whether the recorded effects were specific to the interaction of the resistance proteins with DsbA, and not a result of a general inability of the *dsbA* mutant strain to resist antibiotic stress. We observed no decreases in MIC values for the aminoglycoside antibiotic gentamicin, which is not affected by the activity of the tested enzymes (*Figure 1B*, *Figure 1—figure supplement 6*). Furthermore, the β-lactam MIC values of strains harboring the empty-vector alone, or a plasmid encoding L2-1 (*Figure 1B*), a β-lactamase containing three cysteine residues, but no disulfide bond (PDB ID: 1O7E), remained unchanged. Finally, to rule out the possibility that deletion of *dsbA* caused changes in cell envelope integrity that might confound our results, we measured the permeability of the outer and inner membrane of the *dsbA* mutant. To assess the permeability of the outer membrane, we used the fluorescent dye 1-N-phenylnaphthylamine (NPN) and complemented our results with vancomycin MIC assays (*Figure 1—figure supplement 7A*). To test the integrity of the entire cell envelope, we used the fluorescent dye propidium iodide (PI), as well as the β-galactosidase substrate chlorophenyl red-β-D-galactopyranoside (CPRG) (*Figure 1—figure supplement 7B*). All four assays confirmed that the cell envelope integrity of the *dsbA* mutant is comparable to the parental strain (*Figure 1—figure supplement 7*). Together, these results indicate that many cell envelope AMR determinants that contain more than one cysteine residue are substrates of DsbA and that the process of disulfide bond formation is important for their activity.

Unlike β-lactamases and MCR enzymes, none of the components of the six *E. coli* RND efflux pumps contain periplasmic cysteine residues (*Wang et al., 2017*), and thus they are not substrates of the DSB system. Nonetheless, as DsbA assists the folding of approximately 300 extracytoplasmic proteins, and plays a central role in maintaining the homeostasis of the cell envelope proteome (*Kadokura et al., 2004*; *Dutton et al., 2008*; *Vertommen et al., 2008*), we wanted to assess whether changes in periplasmic proteostasis that occur in its absence could indirectly influence efflux pump function. To do this, we determined the MIC values of three antibiotics that are RND efflux pump substrates using *E. coli* MG1655, a model strain for efflux studies, its *dsbA* mutant, and a mutant lacking *acrA*, an essential component of the major *E. coli* RND pump AcrAB-TolC. MIC values for the *dsbA* mutant were lower than for the parental strain for all tested substrate antibiotics, but remained unchanged for the non-substrate gentamicin (*Figure 1D*). This indicates that the MG1655 *dsbA* strain is generally able to resist antibiotic stress as efficiently as its parent, and that the recorded decreases in MIC values are specific to efflux pump function in the absence of DsbA. As expected for a gene deletion of a pump component, the *acrA* mutant had substantially lower MIC values for effluxed antibiotics (*Figure 1D*). At the same time, even though gentamicin is not effluxed by AcrAB-TolC (*Nikaido, 1996*), the gentamicin MIC of the *acrA* mutant was twofold lower than that of *E. coli* MG1655, in agreement with the fact that one of the minor RND pumps in *E. coli*, the aminoglycoside pump AcrD, is entirely reliant on AcrA for its function (*Aires and Nikaido, 2005*; *Rosenberg et al., 2000*; *Yamasaki et al., 2011*). As before, the observed phenotype could be reversed by complementation of *dsbA* (*Figure 1—figure supplement 8*) and the recorded effects were not due to changes in membrane permeability (*Figure 1—figure supplement 9*). Chloramphenicol is the only antibiotic from the tested efflux pump substrates that has a EUCAST breakpoint for Gram-negative bacteria (*E. coli* strains with an MIC of 8 µg/mL or below are classified as sensitive). It is notable that the MIC drop for this pump substrate, caused by deletion of *dsbA*, sensitized the *E. coli* MG1655 *dsbA* strain to chloramphenicol (*Figure 1D*).

Overall, the effect of DsbA absence on efflux pump efficiency is modest and much less substantial than that measured for a mutant lacking *acrA* (2–3-fold decrease in MIC versus 5–16-fold decrease, respectively) (*Figure 1D*). Nonetheless, the recorded decreases in MIC values are robust (*Figure 1D*) and in agreement with previous studies reporting that deletion of *dsbA* increases the sensitivity of *E. coli* to dyes like acridine orange and pyronin Y (*Bardwell et al., 1991*), which are known substrates of AcrAB-TolC. While it is unlikely that the decreases in MIC values for effluxed antibiotics in the absence of DsbA are of clinical significance, it is interesting to explore the mechanistic relationship between DsbA and efflux pumps further, because there are very few examples of DsbA being important for the function of extra-cytoplasmic proteins independent from its disulfide bond forming capacity (*Alonso-Caballero et al., 2018*; *Zheng et al., 1997*).

## Altered periplasmic proteostasis due to the absence of DsbA results in degradation or misfolding of cysteine-containing resistance determinants and sub-optimal function of efflux pumps

To understand the underlying mechanisms that result in the decreased MIC values observed for the *dsbA* mutant strains, we assessed the protein levels of a representative subset of β-lactamases (GES-1, L1-1, KPC-3, FRI-1, OXA-4, OXA-10, OXA-198) and all tested MCR enzymes by immunoblotting. When expressed in the *dsbA* mutant, all Ambler class A and B β-lactamases (*Table 1*), except GES-1 which we were not able to visualize by immunoblotting, exhibited drastically reduced protein levels whilst the amount of the control enzyme L2-1 remained unaffected (*Figure 2A*). This suggests that when these enzymes lack their disulfide bond, they are ultimately degraded. We did not detect any decrease in protein amounts for Ambler class D enzymes (*Table 1*, *Figure 2B*). However, the hydrolytic activity of these β-lactamases was significantly lower in the *dsbA* mutant (*Figure 2C*), suggesting a folding defect that leads to loss of function.

Like with class A and B β-lactamases, MCR enzymes were undetectable when expressed in a *dsbA* mutant (*Figure 3A*) suggesting that their stability or folding is severely compromised when they lack their disulfide bonds. We further confirmed this by directly monitoring the lipid A profile of all MCR-expressing strains where deletion of *dsbA* resulted in colistin MIC values of 2 µg/mL or lower (i.e. strains expressing MCR-3, –4, –5, and –8, *Figure 1C*) using MALDI-TOF mass spectrometry (*Figure 3BC*). MCR activity leads to the addition of phosphoethanolamine to the lipid A portion of bacterial lipopolysaccharide (LPS), resulting in reduced binding of colistin to LPS and, thus, resistance.

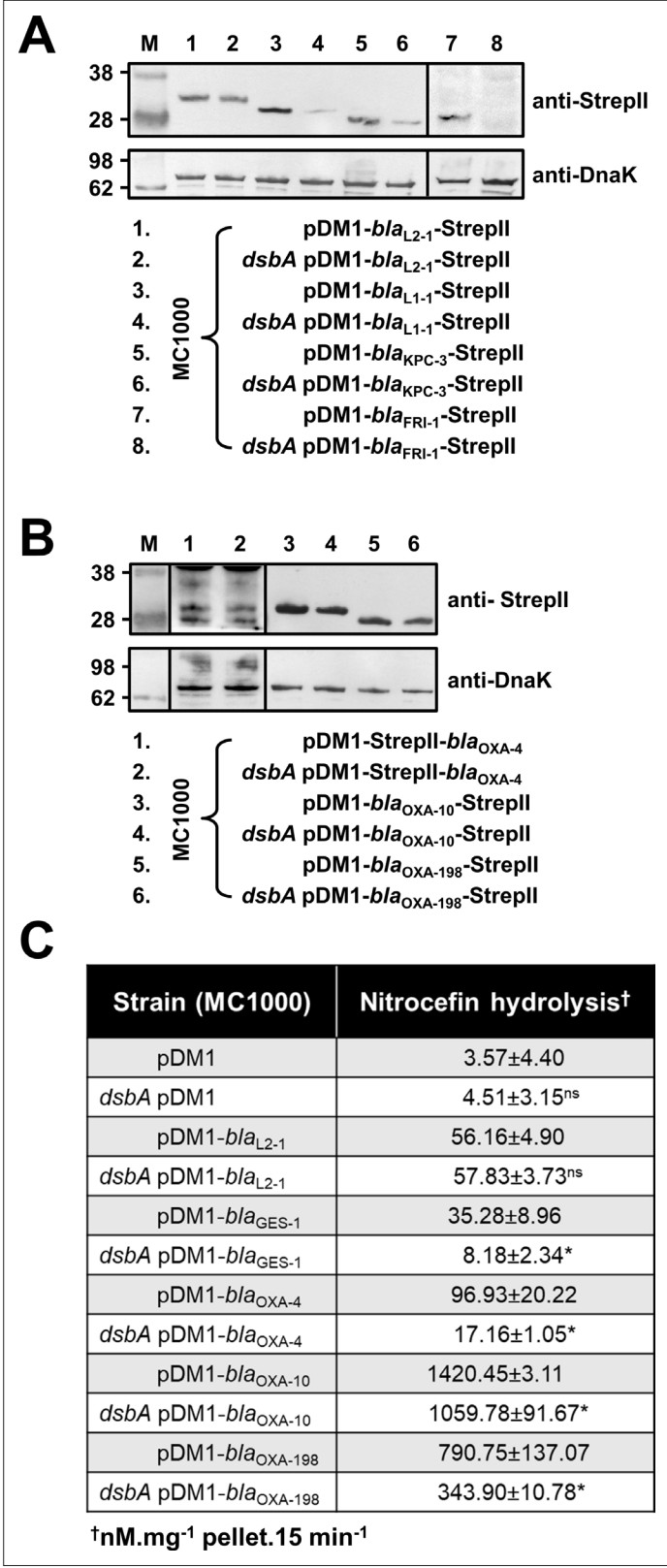

**Figure 2.** β-lactamase enzymes from most classes become unstable in the absence of DsbA. (**A**) Protein levels of disulfide-bond-containing Ambler class A and B β-lactamases are drastically reduced when these enzymes are expressed in *E. coli* MC1000 *dsbA*; the amount of the control enzyme L2-1 is unaffected. (**B**) Protein levels of Class D disulfide-bond-containing β-lactamases are unaffected by the absence of DsbA. OXA-4 is detected as two bands

*Figure 2 continued on next page*

*Figure 2 continued*

at ~28 kDa. For panels (**A**) and (**B**) protein levels of StrepII-tagged β-lactamases were assessed using a Strep-Tactin-AP conjugate or a Strep-Tactin-HRP conjugate. A representative blot from three biological experiments, each conducted as a single technical repeat, is shown; molecular weight markers (**M**) are on the left, DnaK was used as a loading control and solid black lines indicate where the membrane was cut. (**C**) The hydrolytic activities of the tested Class D β-lactamases and of the Class A enzyme GES-1, which could not be detected by immunoblotting, are significantly reduced in the absence of DsbA. The hydrolytic activities of strains harboring the empty vector or expressing the control enzyme L2-1 show no dependence on DsbA. n = 3 (each conducted in technical duplicate), table shows means ± SD, significance is indicated by * = p < 0.05, ns = non-significant.

The online version of this article includes the following source data for figure 2:

**Source data 1.** Original files of the full raw unedited immunoblots used to prepare *Figure 2A*.

**Source data 2.** Uncropped immunoblots used to prepare *Figure 2A*.

**Source data 3.** Original files of the full raw unedited immunoblots used to prepare *Figure 2B*.

**Source data 4.** Uncropped immunoblots used to prepare *Figure 2B*.

---

In *E. coli,* the major lipid A peak detected by mass spectrometry is present at *m/z* 1796.2 (*Figure 3B*, first spectrum) and it corresponds to hexa-acyl diphosphoryl lipid A (native lipid A). The lipid A profile of *E. coli* MC1000 *dsbA* was identical to that of the parental strain (*Figure 3B*, second spectrum). In the presence of MCR enzymes two additional peaks were observed, at *m/z* 1821.2 and 1919.2 (*Figure 3B*, third spectrum). The peak at *m/z* 1919.2 corresponds to the addition of a phosphoethanolamine moiety to the phosphate group at position 1 of native lipid A, and the peak at *m/z* 1821.2 corresponds to the addition of a phosphoethanolamine moiety to the 4′ phosphate of native lipid A and the concomitant loss of the phosphate group at position 1 (*Dortet et al., 2018*). For *dsbA* mutants expressing MCR-3, –5, and –8 (*Figure 3C*), the peaks at *m/z* 1821.2 and *m/z* 1919.2 could no longer be detected, whilst the native lipid A peak at *m/z* 1796.2 remained unchanged (*Figure 3B*, fourth spectrum); *dsbA* mutants expressing MCR-4 retain some basal lipid A-modifying activity, nonetheless this is not sufficient for this strain to efficiently evade colistin treatment (*Figure 1C*). Together these data suggest that in the absence of DsbA, MCR enzymes are unstable (*Figure 3A*) and therefore no longer able to efficiently catalyze the addition of phosphoethanolamine to native lipid A (*Figure 3BC*); as a result, they cannot confer resistance to colistin (*Figure 1C*).

As RND efflux pump proteins do not contain any disulfide bonds, the decreases in MIC values for pump substrates in the absence of *dsbA* (*Figure 1D*) are likely mediated by additional cell-envelope components. The protease DegP, previously found to be a DsbA substrate (*Hiniker and Bardwell, 2004*), seemed a promising candidate for linking DsbA to efflux pump function. DegP degrades a range of misfolded extracytoplasmic proteins including, but not limited to, subunits of higher order protein complexes and proteins lacking their native disulfide bonds (*Clausen et al., 2002*). We hypothesized that in a *dsbA* mutant the substrate burden on DegP would be dramatically increased, whilst DegP itself would not function optimally due to absence of its disulfide bond (*Skórko-Glonek et al., 2003*). Consequently, protein turn over in the cell envelope would not occur efficiently. Since the essential RND efflux pump component AcrA needs to be cleared by DegP when it becomes misfolded or nonfunctional (*Gerken and Misra, 2004*), we expected that the reduced DegP efficiency in a *dsbA* mutant would result in accumulation of nonfunctional AcrA in the periplasm, which would then interfere with pump function. In agreement with our hypothesis, we found that in the absence of DsbA degradation of DegP occurred (*Figure 4A*), reducing the pool of active enzyme (*Skórko-Glonek et al., 2003*). In addition, AcrA accumulated to the same extent in a *dsbA* and a *degP* mutant (*Figure 4B*), suggesting that in both these strains AcrA was not efficiently cleared. Finally, no accumulation was detected for the outer-membrane protein TolC (*Figure 4C*), which is not a DegP substrate (*Werner et al., 2003*). Thus, in the absence of DsbA, inefficient DegP-mediated periplasmic proteostasis affects RND efflux pumps (*Figure 1D*) through the accumulation of AcrA that should have been degraded and removed from the cell envelope.

The data presented above validate our initial hypothesis. The absence of DsbA affects the stability and folding of cysteine-containing resistance proteins and in most cases leads to drastically reduced protein levels for the tested enzymes. As a result, and in agreement with the recorded decreases in MIC values (*Figure 1BC*), these folding defects impede the ability of AMR determinants that are substrates of DsbA to confer resistance (*Figure 4D*). In addition, changes in cell envelope protein

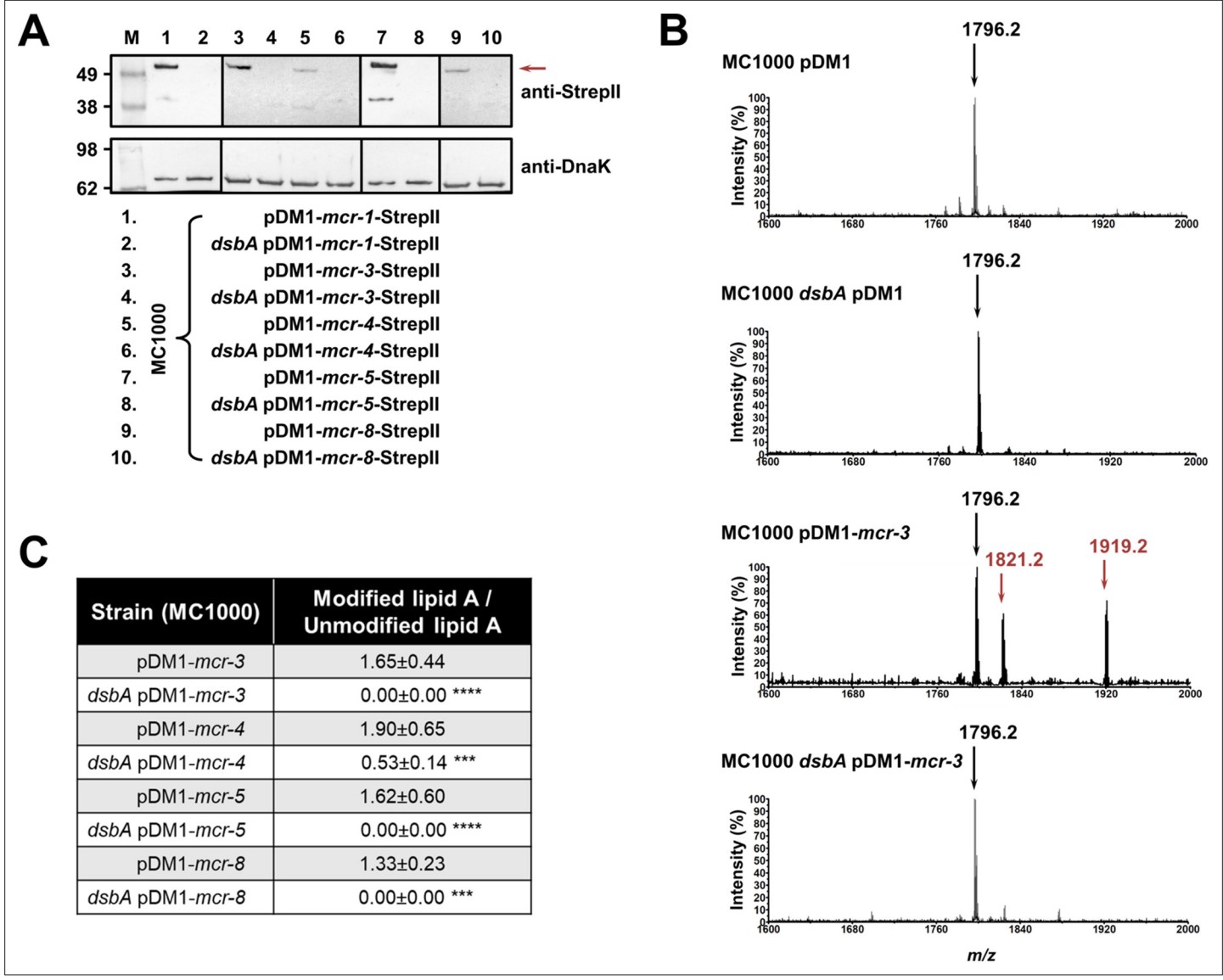

**Figure 3.** MCR enzymes become unstable in the absence of DsbA. (**A**) The amounts of MCR proteins are drastically reduced when they are expressed in *E. coli* MC1000 *dsbA*; the red arrow indicates the position of the MCR-specific bands. Protein levels of StrepII-tagged MCR enzymes were assessed using a Strep-Tactin-AP conjugate. A representative blot from three biological experiments, each conducted as a single technical repeat, is shown; molecular weight markers (M) are on the left, DnaK was used as a loading control and solid black lines indicate where the membrane was cut. (**B**) The ability of MCR enzymes to transfer phoshoethanolamine to the lipid A portion of LPS is either entirely abrogated or significantly reduced in the absence of DsbA. This panel shows representative MALDI-TOF mass spectra of unmodified and MCR-modified lipid A in the presence and absence of DsbA. In *E. coli,* MC1000 and MC1000 *dsbA* the major peak for native lipid A is detected at *m/z* 1796.2 (first and second spectrum, respectively). In the presence of MCR enzymes (*E. coli* MC1000 expressing MCR-3 is shown as a representative example), two additional peaks are observed, at *m/z* 1821.2 and 1919.2 (third spectrum). For *dsbA* mutants expressing MCR enzymes (*E. coli* MC1000 *dsbA* expressing MCR-3 is shown), these additional peaks are not present, whilst the native lipid A peak at *m/z* 1796.2 remains unchanged (fourth spectrum). Mass spectra are representative of the data generated from four biological experiments, each conducted as a technical duplicate. (**C**) Quantification of the intensities of the lipid A peaks recorded by MALDI-TOF mass spectrometry for all tested MCR-expressing strains. n = 4 (each conducted in technical duplicate), table shows means ± SD, significance is indicated by *** = p < 0.001 or **** = p < 0.0001.

The online version of this article includes the following source data for figure 3:

**Source data 1.** Original files of the full raw unedited immunoblots used to prepare *Figure 3A* for which a Strep-Tactin-AP conjugate and an anti-DnaK 8E2/2 antibody were used.

**Source data 2.** Uncropped immunoblots used to prepare *Figure 3A*.

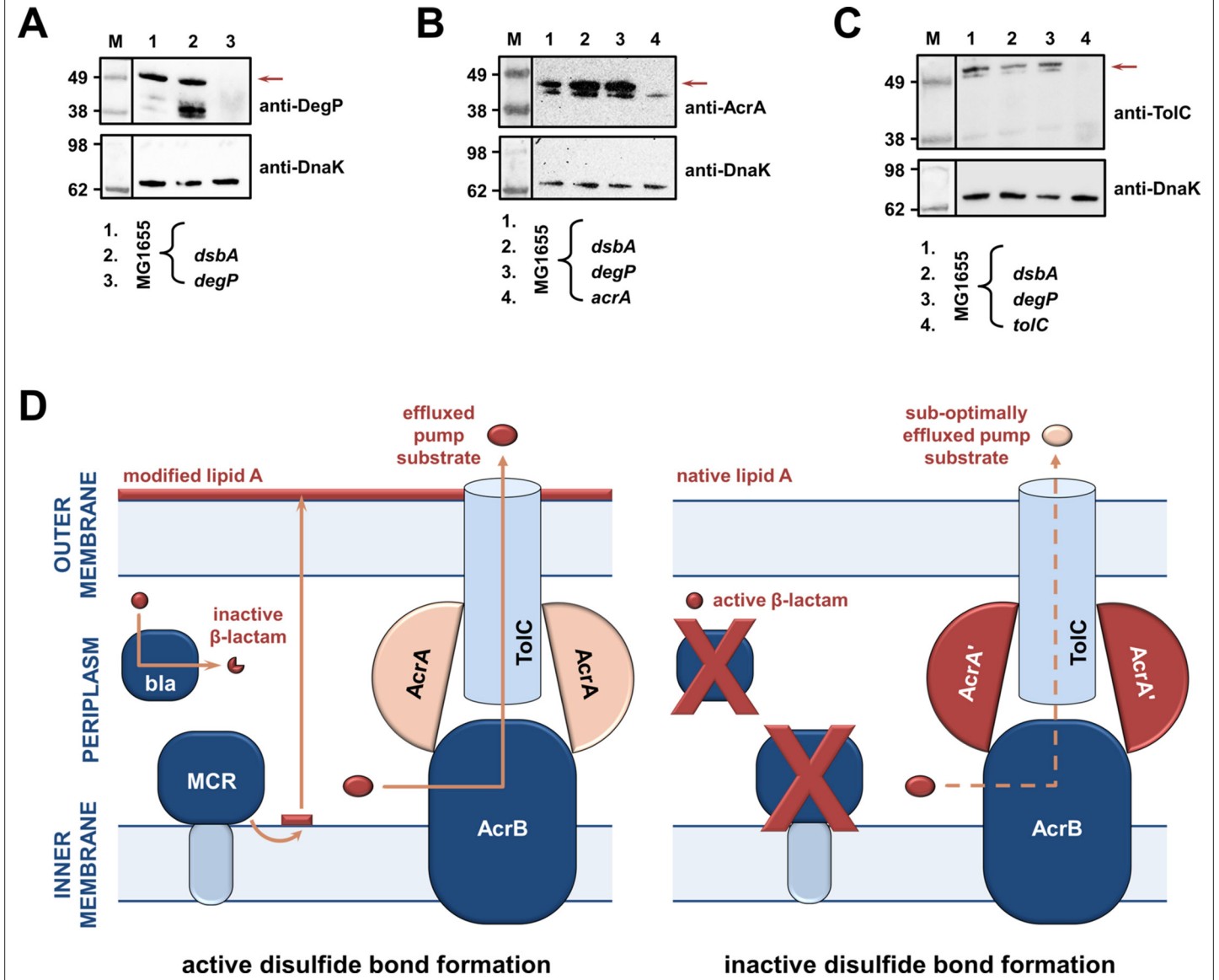

**Figure 4.** RND efflux pump function is impaired in the absence of DsbA due to accumulation of unfolded AcrA resulting from insufficient DegP activity (**A, B, C**) . (**A**) In the absence of DsbA the pool of active DegP is reduced. In *E. coli* MG1655 (lane 1), DegP is detected as a single band, corresponding to the intact active enzyme. In *E. coli* MG1655 *dsbA* (lane 2), an additional lower molecular weight band of equal intensity is present, indicating that DegP is degraded in the absence of its disulfide bond (*Hiniker and Bardwell, 2004*; *Skórko-Glonek et al., 2003*). DegP protein levels were assessed using an anti-DegP primary antibody and an HRP-conjugated secondary antibody. *E. coli* MG1655 *degP* was used as a negative control for DegP detection (lane 3); the red arrow indicates the position of intact DegP. (**B**) The RND pump component AcrA accumulates to the same extent in the *E. coli* MG1655 *dsbA* and *degP* strains, indicating that in both strains protein clearance is affected. AcrA protein levels were assessed using an anti-AcrA primary antibody and an HRP-conjugated secondary antibody. *E. coli* MG1655 *acrA* was used as a negative control for AcrA detection; the red arrow indicates the position of the AcrA band. (**C**) TolC, the outer-membrane channel of the AcrAB pump, does not accumulate in a *dsbA* or a *degP* mutant. TolC is not a DegP substrate (*Werner et al., 2003*), hence similar TolC protein levels are detected in *E. coli* MG1655 (lane 1) and its *dsbA* (lane 2) and *degP* (lane 3) mutants. TolC protein levels were assessed using an anti-TolC primary antibody and an HRP-conjugated secondary antibody. *E. coli* MG1655 *tolC* was used as a negative control for TolC detection (lane 4); the red arrow indicates the position of the bands originating from TolC. For all panels a representative blot from three biological experiments, each conducted as a single technical repeat, is shown; molecular weight markers (M) are on the left, DnaK was used as a loading control and solid black lines indicate where the membrane was cut. (**D**) Impairing disulfide bond formation in the cell envelope simultaneously affects distinct AMR determinants. (Left) When DsbA is present, that is, when disulfide bond formation occurs, degradation of β-lactam antibiotics by β-lactamases (marked 'bla'), modification of lipid A by MCR proteins and active efflux of RND pump substrates lead to resistance. The major *E. coli* RND efflux pump AcrAB-TolC is depicted in this schematic as a characteristic example. (**Right**) In the asucess of disulfide bond formation is impaired, most cysteine-containing β-lactamases as well as MCR proteins are unstable and degrade, making bacteria susceptible to

*Figure 4 continued on next page*

*Figure 4 continued*

β-lactams and colistin, respectively. Absence of DsbA has also a general effect on proteostasis in the cell envelope which results in reduced clearance of nonfunctional AcrA-like proteins (termed 'AcrA' and depicted in dark red color) by periplasmic proteases. Insufficient clearance of these damaged AcrA components from the pump complex makes efflux less efficient.

The online version of this article includes the following source data for figure 4:

**Source data 1.** Original files of the full raw unedited immunoblots used to prepare *Figure 4A*.

**Source data 2.** Uncropped immunoblots used to prepare *Figure 4A*.

**Source data 3.** Original files of the full raw unedited immunoblots used to prepare *Figure 4B*.

**Source data 4.** Uncropped immunoblots used to prepare *Figure 4B*.

**Source data 5.** Original files of the full raw unedited immunoblots used to prepare *Figure 4C*.

**Source data 6.** Uncropped immunoblots used to prepare *Figure 4C*.

homeostasis due to the lack of DSB activity can result in a generalized, albeit much more modest, effect on protein function in this compartment. This is suggested by the fact that prevention of disulfide bond formation seems to indirectly affect the AcrAB-TolC efflux pump (*Figure 1D*), because of insufficient turnover of its AcrA component (*Figure 4D*).

## Sensitization of clinical isolates to existing antibiotics can be achieved by chemical inhibition of DsbA activity

DsbA is essential for the folding of many virulence factors. As such, inhibition of the DSB system has been proposed as a promising anti-virulence strategy (*Heras et al., 2009*; *Landeta et al., 2018*; *Heras et al., 2015*) and efforts have been made to develop inhibitors for DsbA (*Duprez et al., 2015*; *Totsika et al., 2018*), its redox partner DsbB (*Figure 1A*; *Landeta et al., 2015*) or both (*Halili et al., 2015*). These studies have made the first steps toward the production of chemical compounds that inhibit the function of the DSB proteins, providing us with a laboratory tool to test our approach against AMR.

4,5-Dichloro-2-(2-chlorobenzyl)pyridazin-3-one, termed 'compound 12' in Landeta et al. (*Landeta et al., 2015*) is a potent laboratory inhibitor of *E. coli* DsbB and its analogues from closely related organisms. Using this molecule, we could chemically inhibit the function of the DSB system. We first tested the motility of *E. coli* MC1000 in the presence of the inhibitor and found that cells were significantly less motile (*Figure 5AB*), consistent with the fact that impairing DSB function prevents the formation of the flagellar P-ring component FlgI (*Dailey and Berg, 1993*; *Hizukuri et al., 2006*). Furthermore, we directly assessed the redox state of DsbA in the presence of 'compound 12' to probe whether it was being re-oxidized by DsbB, a necessary step that occurs after each round of oxidative protein folding and allows DsbA to remain active (*Figure 1A*). Under normal growth conditions, DsbA was in its active oxidized form in the bacterial periplasm (i.e. C30 and C33 form a disulfide bond), showing that it was efficiently regenerated by DsbB (*Kishigami et al., 1995*; *Figure 5C*). By contrast, addition of the inhibitor to growing *E. coli* MC1000 cells resulted in accumulation of inactive reduced DsbA, thus confirming that DsbB function was impeded (*Figure 5C*).

After testing the efficacy of the DsbB inhibitor, we proceeded to examine whether chemical inhibition of the DSB system could be used to broadly impair the function of AMR determinants. We determined MIC values for the latest generation β-lactam that each β-lactamase can hydrolyze, or colistin, for our panel of *E. coli* MC1000 strains and found that addition of the compound during MIC testing phenocopied the effects of a *dsbA* deletion on β-lactamase and MCR activity (*Figure 5DE*, *Figure 5—figure supplement 1*, *Supplementary file 2b*). The observed effects are not a result of altered cell growth, as addition of the compound does not affect the growth profile of the bacteria (*Figure 5—figure supplement 2A*), in agreement with the fact that deletion of *dsbA* does not affect cell viability (*Figure 5—figure supplement 2B*). Furthermore, the changes in the recorded MIC values are due solely to inhibition of the DSB system as no additive effects on MIC values were observed when the *dsbA* mutant harboring a β-lactamase or *mcr* gene was exposed to the compound (*Figure 5—figure supplement 3*).

Having shown that the DSB system is a tractable target in the context of AMR, we examined the effect of chemical inhibition on several species of β-lactamase-expressing Enterobacteria (Supplementary Table 1 in *Supplementary file 3*). We chose to test organisms that pose significant clinical or societal challenges, such as the ESKAPE pathogens *Klebsiella pneumoniae* and *Enterobacter cloacae*

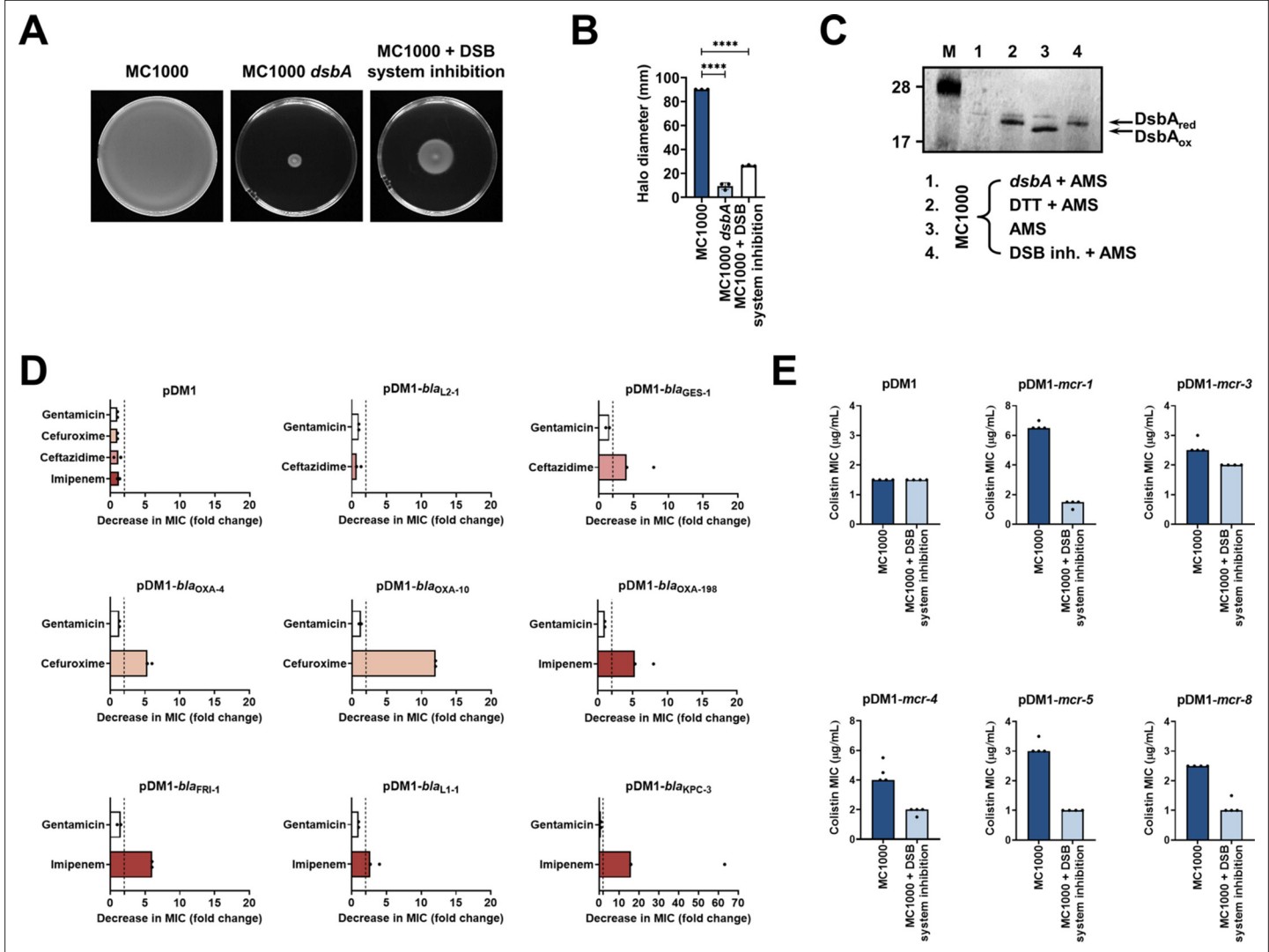

**Figure 5.** Chemical inhibition of the DSB system impedes DsbA function in *E. coli* MC1000 and phenocopies the β-lactam and colistin MIC changes that were observed using a *dsbA* mutant. (**A**) Chemical inhibition of the DSB system impedes flagellar motility in *E. coli* MC1000. A functional DSB system is necessary for flagellar motility in *E. coli* because folding of the P-ring component FlgI requires DsbA-mediated disulfide bond formation (***Dailey and Berg, 1993***). In the absence of DsbA, or upon addition of a chemical inhibitor of the DSB system, the motility of *E. coli* MC1000 is significantly impeded. Representative images of motility plates are shown. (**B**) Quantification of the growth halo diameters in the motility assays shown in panel (**A**). n = 3 (each conducted as a single technical repeat), graph shows means ± SD, significance is indicated by **** = p < 0.0001. (**C**) Chemical inhibition of the DSB system impedes DsbA re-oxidation in *E. coli* MC1000. Addition of the reducing agent DTT to *E. coli* MC1000 bacterial lysates allows the detection of DsbA in its reduced form (DsbA$_{red}$) during immunoblotting; this redox state of the protein, when labeled with the cysteine-reactive compound AMS, shows a 1 kDa size difference (lane 2) compared to oxidized DsbA as found in AMS-labeled but not reduced lysates of *E. coli* MC1000 (lane 3). Addition of a small-molecule inhibitor of DsbB to growing *E. coli* MC1000 cells also results in accumulation of reduced DsbA (lane 4). *E. coli* MC1000 *dsbA* was used as a negative control for DsbA detection (lane 1). A representative blot from two biological experiments, each conducted as a single technical repeat, is shown; DsbA was visualized using an anti-DsbA primary antibody and an AP-conjugated secondary antibody. Molecular weight markers (M) are shown on the left. (**D**) MIC experiments using representative β-lactam antibiotics show that chemical inhibition of the DSB system reduces the MIC values for *E. coli* MC1000 expressing disulfide-bond-containing β-lactamases in a similar manner to the deletion of *dsbA* (compare with ***Figure 1B***). Graphs show MIC fold changes (i.e. MC1000 MIC (µg/mL) / MC1000 + DSB system inhibitor MIC (µg/mL)) for β-lactamase-expressing *E. coli* MC1000 with and without addition of a DSB system inhibitor to the culture medium from two biological experiments, each conducted as a single technical repeat. Black dotted lines indicate an MIC fold change of 2. The aminoglycoside antibiotic gentamicin serves as a control for all strains; gentamicin MIC values (white bars) are unaffected by chemical inhibition of the DSB system (MIC fold changes: < 2). No changes in MIC values (MIC fold changes: < 2) are observed for strains harboring the empty vector control (pDM1) or expressing the class A β-lactamase L2-1, which contains three cysteines but no disulfide bond (PDB ID: 1O7E) (top row). The MIC values used to generate this panel are presented in ***Supplementary file 2b***. (**E**) Colistin MIC experiments show that chemical inhibition of the DSB system reduces the MIC values for *E. coli* MC1000 expressing MCR enzymes in a similar manner to the deletion of *dsbA* (compare with ***Figure 1C***). Colistin MIC values for strains harboring the empty vector control (pDM1) are unaffected by chemical

*Figure 5 continued on next page*

*Figure 5 continued*

inhibition of the DSB system. Graphs show MIC values (µg/mL) from four biological experiments, each conducted in technical quadruplicate, to demonstrate the robustness of the observed effects.

The online version of this article includes the following source data and figure supplement(s) for figure 5:

**Source data 1.** Original file of the full raw unedited immunoblot used to prepare *Figure 5C*, for which an anti-DsbA antibody was used.

**Source data 2.** Uncropped immunoblot used to prepare *Figure 5C*.

**Figure supplement 1.** Gentamicin MIC values for *E. coli* MC1000 strains expressing MCR enzymes.

**Figure supplement 2.** Chemical inhibition of the DSB system or deletion of *dsbA* does not compromise the growth of *E. coli* MC1000.

**Figure supplement 3.** Changes in MIC values observed using the DSB system inhibitor are due solely to inhibition of the DSB system.

(*Mulani et al., 2019*), or drug-resistant *E. coli* strains, which account for 50% of the economic burden of resistant infections (*O'Neill, 2014*). DSB system inhibition in a clinical isolate of *K. pneumoniae* expressing KPC-2 sensitized the strain to imipenem as defined by EUCAST breakpoints (*Figure 6A*). The efficiency of this double treatment is evident from scanning electron micrographs of the tested strains (*Figure 6B*). Addition of either the DSB system inhibitor or imipenem alone does not cause any

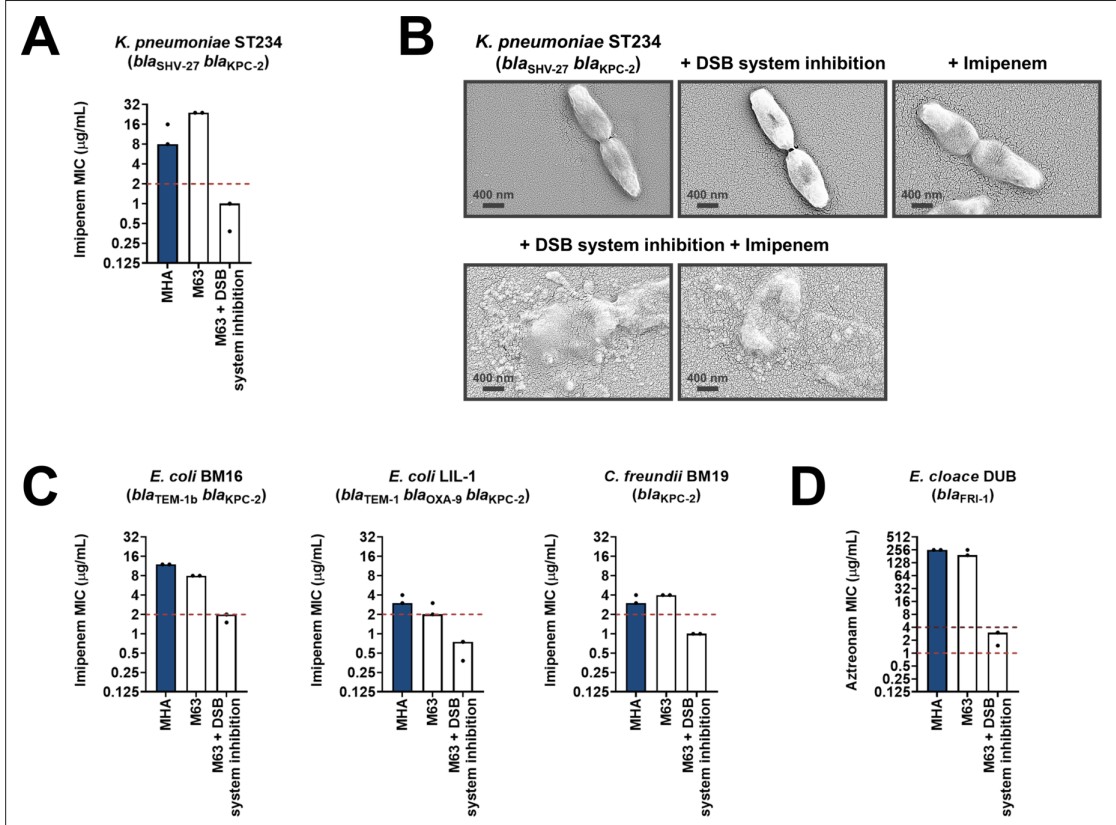

**Figure 6.** Chemical inhibition of the DSB system sensitizes multidrug-resistant clinical isolates to currently available β-lactam antibiotics. (**A**) Addition of a small-molecule inhibitor of DsbB results in sensitization of a *K. pneumoniae* clinical isolate to imipenem. (**B**) Chemical inhibition of the DSB system in the presence of imipenem (final concentration of 6 µg/mL) results in drastic changes in cell morphology for the *K. pneumoniae* clinical isolate used in panel (**A**), while bacteria remain unaffected by single treatments (DSB inhibitor or imipenem). Images show representative scanning electron micrographs of untreated cells (top row, left), cells treated with the DSB inhibitor (top row, middle), cells treated with imipenem (top row, right), and cells treated with both the DSB inhibitor and imipenem (bottom row). Scale bars are at 400 nm. (**C**) Addition of a small-molecule inhibitor of DsbB results in sensitization of *E. coli* and *C. freundii* clinical isolates to imipenem. (**D**) Chemical inhibition of the DSB system of an *E. cloacae* clinical isolate harboring *bla*FRI-1 results in reduction of the aztreonam MIC value by over 180 µg/mL, resulting in intermediate resistance as defined by EUCAST. For panels (**A**), (**C**) and (**D**) graphs show MIC values (µg/ml) from two biological experiments, each conducted as a single technical repeat. MIC values determined using Mueller-Hinton agar (MHA) in accordance with the EUCAST guidelines (dark blue bars) are comparable to the values obtained using defined media (M63 agar, white bars); use of growth media lacking small-molecule oxidants is required for the DSB system inhibitor to be effective. Red dotted lines indicate the EUCAST clinical breakpoint for each antibiotic, and purple dotted lines indicate the EUCAST threshold for intermediate resistance.

changes in the morphology of *K. pneumoniae* cells, which remain healthy and dividing (*Figure 6B*, top row). By contrast, the combination of the inhibitor with imipenem (added at a sub-MIC final concentration of 6 μg/mL), led to dramatic changes in the appearance of the cells, whose integrity was entirely compromised (*Figure 6B*, bottom row). Similarly, *E. coli* and *Citrobacter freundii* isolates expressing KPC-2, including multidrug-resistant strains, also showed clinically relevant decreases in their MIC values for imipenem that resulted in sensitization when their DSB system was chemically inhibited (*Figure 6C*). For an *E. cloacae* isolate expressing FRI-1, chemical inhibition of DsbA caused reduction in its aztreonam MIC value by over 180 μg/mL, resulting in intermediate resistance as defined by EUCAST breakpoints (*Figure 6D*).

Along with β-lactamase-expressing strains, we also tested the effect of DsbA inhibition on MCR-producing clinical isolates. We found that combination of the DSB system inhibitor with colistin led to reduction of the colistin MIC and sensitization of MCR-1-expressing multidrug-resistant *E. coli* (*Figure 7A*). In agreement with this, SEM images of this strain after combination treatment using sub-MIC amounts of colistin (final concentration of 2 μg/mL) revealed drastic changes in morphology, whereby cells blebbed intensely or their contents leaked out (*Figure 7B*). We tested eight additional clinical *E. coli* isolates that encode diverse MCR enzymes (most of which are multidrug resistant) and have colistin MICs ranging from 3 to 16 μg/mL; DSB system inhibition also allowed sensitization to colistin (*Figure 7C*) for tested strains. At the same time, we were able to show that DSB system inhibition in *E. coli* CNR1790 (i.e. the clinical isolate expressing both MCR-1 and the ESBL TEM-15 that was sensitized to colistin in *Figure 7A*), led to a decrease in its ceftazidime MIC, resulting in intermediate resistance (*Figure 7D*). While we did not test the dependence of TEM enzymes on DsbA in our panel of *E. coli* K-12 strains, we chose to test the effects of DSB system inhibition on *E. coli* CNR1790 because we posited that the disulfide bond in TEM-15 may be important for its function, based on the fact that the narrow-spectrum TEM-1 enzyme has been shown to be reliant on its disulfide under stress conditions (*Schultz et al., 1987*). Validation of our hypothesis provides evidence that DsbA inhibition can improve the resistance profile of the same isolate both for β-lactam (*Figure 7D*) and polymyxin (*Figure 7A*) antibiotics. Together these results, obtained using multiple clinical strains from several bacterial species, provide further proof of the significance of our data from heterologously expressed β-lactamase and MCR enzymes in *E. coli* K-12 strains (*Figure 1BC*), and showcase the potential of this approach for clinical applications.

To determine if our approach for Enterobacteria would be appropriate for other multidrug-resistant Gram-negative bacteria, we tested it on another major ESKAPE pathogen, *Pseudomonas aeruginosa* (*Mulani et al., 2019*). This bacterium has two DsbB analogues which are functionally redundant (*Arts et al., 2013*). The chemical inhibitor used in this study has been shown to be effective against DsbB1, but much less effective against DsbB2 of *P. aeruginosa* PA14 (*Landeta et al., 2015*), making it unsuitable for MIC assays on *P. aeruginosa* clinical isolates. Nonetheless, deletion of *dsbA1* in a multidrug-resistant *P. aeruginosa* clinical isolate expressing OXA-198 (PA43417), led to sensitization of this strain to the antipseudomonal β-lactam piperacillin (*Figure 8A*). In addition, we deleted *dsbA1* in the multidrug-resistant *P. aeruginosa* PAe191 strain that produces OXA-19, a member of the OXA-10 phylogenetic family (*Supplementary file 1*) and the most disseminated OXA enzyme in clinical strains (*Mugnier et al., 1998*). In this case, absence of DsbA caused a drastic reduction in the ceftazidime MIC value by over 220 μg/mL, and sensitized the strain to aztreonam (*Figure 8B*). These results suggest that targeting disulfide bond formation could be useful for the sensitization of many more clinically important Gram-negative species.

Finally, to test our approach in an infection context we performed in vivo survival assays using the wax moth model *Galleria mellonella* (*Figure 8C*). *G. mellonella* has proven to be an invaluable non-vertebrate model for the study of *P. aeruginosa* pathogenesis as well as for testing antibiotic treatments against this organism (*Hill et al., 2014*; *Miyata et al., 2003*), making it an appropriate tool for assessing the in vivo efficacy of our approach on a multidrug-resistant strain of this pathogen. Larvae were infected with the *P. aeruginosa* PAe191 strain producing OXA-19, and its *dsbA1* mutant, and infections were treated once with ceftazidime at a final concentration below the EUCAST breakpoint. No larvae survived beyond 18 hr post infection with *P. aeruginosa* PAe191, even when treatment with ceftazidime was performed (*Figure 8C*, blue and red survival curves). Deletion of *dsbA1* resulted in 80% mortality of the larvae at 50 hr post infection (*Figure 8C*, light blue survival curve); this increase in survival compared to larvae infected with *P. aeruginosa* PAe191 is due to the fact that absence of the

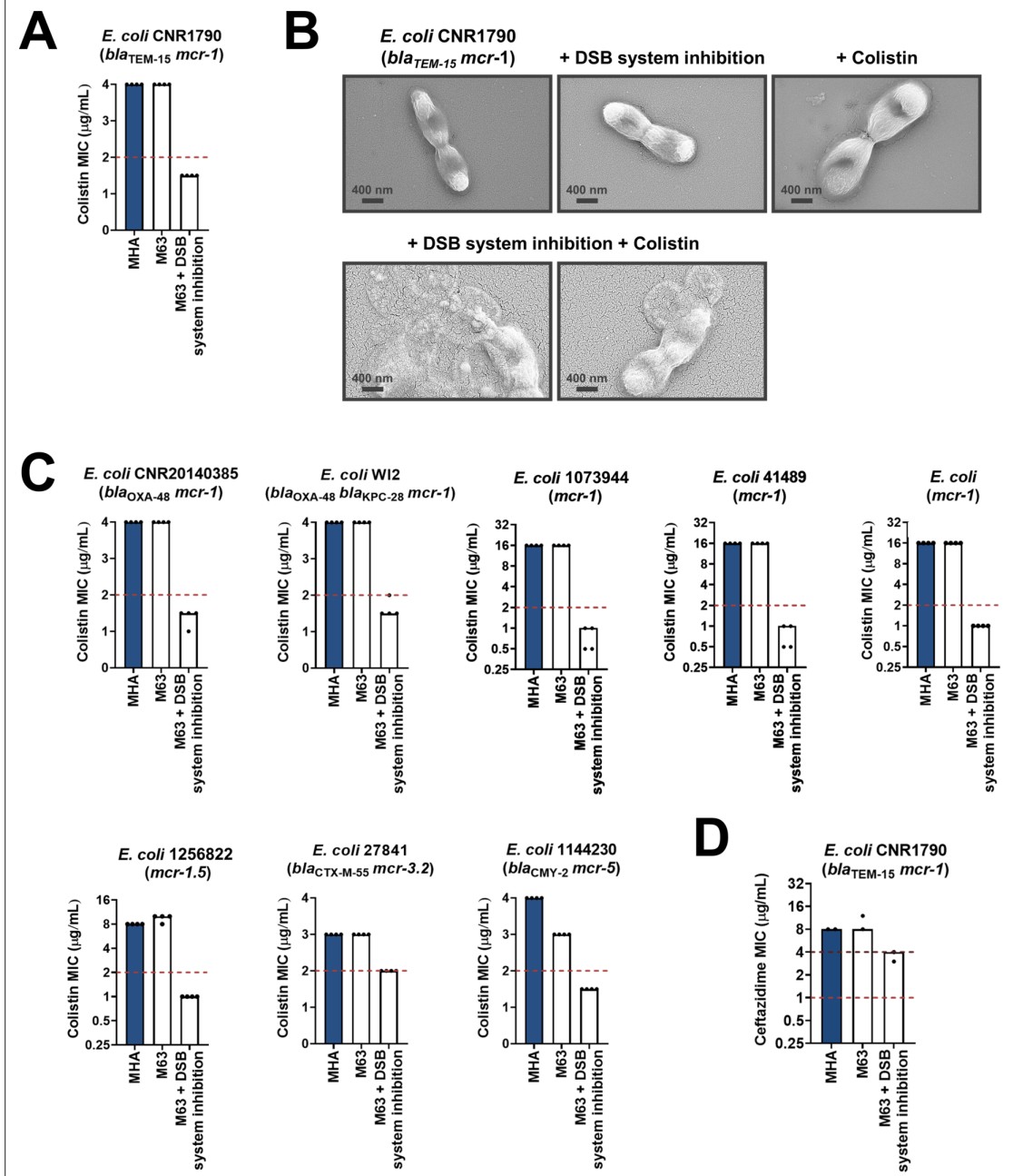

**Figure 7.** Chemical inhibition of the DSB system sensitizes multidrug-resistant clinical isolates to colistin. (**A**) Addition of a small-molecule inhibitor of DsbB to a colistin-resistant clinical *E. coli* isolate expressing MCR-1 results in sensitization to colistin. (**B**) Chemical inhibition of the DSB system in the presence of colistin (final concentration of 2 µg/mL) results in drastic changes in cell morphology for the *E. coli* clinical isolate used in panel (A), while bacteria remain unaffected by single treatments (DSB inhibitor or colistin). Images show representative scanning electron micrographs of untreated cells (top row, left), cells treated with the DSB inhibitor (top row, middle), cells treated with colistin (top row, right), and cells treated with both the DSB inhibitor and colistin (bottom row). Scale bars are at 400 nm. (**C**) Chemical inhibition of the DSB system results in sensitization of four additional colistin-resistant *E. coli* strains expressing MCR enzymes. For panels (**A**) and (**C**), graphs show MIC values (µg/mL) from four biological experiments, each conducted in technical quadruplicate, to demonstrate the robustness of the observed effects. (**D**) Use of the DSB system inhibitor on the same clinical *E. coli* isolate tested in panel (**A**), results in intermediate resistance for ceftazidime as defined by EUCAST. The graph shows MIC values (µg/ml) from two biological experiments, each conducted as a single technical repeat. For panels (**A**), (**C**), (**D**), MIC values determined using Mueller-Hinton agar (MHA) in accordance with the EUCAST guidelines (dark blue bars) are comparable to the values obtained using defined media (M63 agar, white bars); use of growth media lacking small-molecule oxidants is required for the DSB system inhibitor to be effective. For all panels, red dotted lines indicate the EUCAST clinical breakpoint for each antibiotic, and purple dotted lines indicate the EUCAST threshold for intermediate resistance.

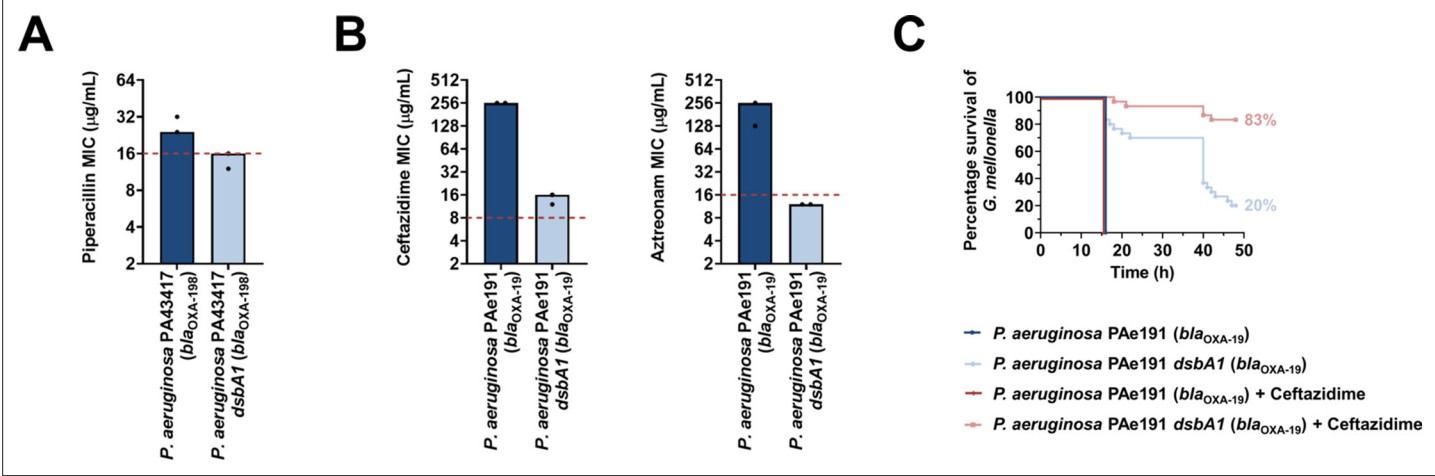

**Figure 8.** Absence of the principal DsbA analogue (DsbA1) from *P. aeruginosa* clinical isolates expressing OXA enzymes sensitizes them to existing β-lactam antibiotics and dramatically increases the survival of infected G. *mellonella* larvae that undergo antibiotic treatment. (**A**) Absence of DsbA1 sensitizes the *P. aeruginosa* PA43417 clinical isolate expressing OXA-198 to the first-line antibiotic piperacillin. (**B**) Absence of DsbA1 sensitizes the *P. aeruginosa* PAe191 clinical isolate expressing OXA-19 to aztreonam and results in reduction of the ceftazidime MIC value by over 220 µg/mL. For panels (**A**) and (**B**) the graphs show MIC values (µg/ml) from two biological experiments, each conducted as a single technical repeat; red dotted lines indicate the EUCAST clinical breakpoint for each antibiotic. (**C**) 100% of the *G. mellonella* larvae infected with *P. aeruginosa* PAe191 (blue curve) or infected with *P. aeruginosa* PAe191 and treated with 7.5 µg/mL ceftazidime (red curve) die 18 hr post infection, and only 20% of the larvae infected with *P. aeruginosa* PAe191 *dsbA1* (light blue curve) survive 50 hr post infection. Treatment of larvae infected with *P. aeruginosa* PAe191 *dsbA1* with 7.5 µg/mL ceftazidime (pink curve) results in 83% survival, 50 hr post infection. The graph shows Kaplan-Meier survival curves of infected *G. mellonella* larvae after different treatment applications; horizontal lines represent the percentage of larvae surviving after application of each treatment at the indicated time point (a total of 30 larvae were used for each curve). Statistical analysis of this data was performed using a Mantel-Cox test; n = 30; p =< 0.0001 (significance) (*P. aeruginosa* versus *P. aeruginosa dsbA1*), p > 0.9999 (non-significance) (*P. aeruginosa* vs *P. aeruginosa* treated with ceftazidime), p =< 0.0001 (significance) (*P. aeruginosa* treated with ceftazidime versus *P. aeruginosa dsbA1*), p =< 0.0001 (significance) (*P. aeruginosa dsbA1* versus *P. aeruginosa dsbA1* treated with ceftazidime).

principal DsbA protein likely affects the virulence of the pathogen (*Landeta et al., 2019*). Nonetheless, treatment of the *dsbA1* mutant with ceftazidime resulted in a significant increase in survival (17% mortality) compared to the untreated condition, 50 hr post infection (*Figure 8C*, compare the light blue and pink survival curves). This improvement in survival is even more noticeable if one compares the survival of larvae treated with ceftazidime after infection with *P. aeruginosa* PAe191 versus infection with *P. aeruginosa* PAe191 *dsbA1* (*Figure 8C*, compare the red and pink survival curves). Since OXA-19, in this case produced by a multi-drug resistant clinical strain (Supplementaty Table 1 in *Supplementary file 3*, *Figure 8B*), is a broad-spectrum β-lactamase that cannot be neutralized by classical β-lactamase inhibitors (*Table 1*), these results further highlight the promise of our approach for future clinical applications.

## Discussion

This work is one of the first reports of a strategy capable of simultaneously impairing multiple types of AMR determinants by compromising the function of a single target. By inhibiting DsbA, a non-essential cell envelope protein which is unique to bacteria, we can inactivate diverse resistance enzymes and sensitize critically important pathogens to several existing antibiotics. This proof of principle will hopefully further incentivize the development of DsbA inhibitors and open new avenues toward the inception of novel adjuvants that will help reverse AMR in Gram-negative organisms.

We have shown that targeting DsbA incapacitates broad-spectrum β-lactamases from three of the four Ambler classes (class A, B and D, *Figure 1B*). This includes enzymes that are not susceptible to classical β-lactamase inhibitors (*Table 1*), such as members of the KPC and OXA families, as well as metallo-β-lactamases like L1-1 from the often pan-resistant organism *Stenotrophomonas maltophilia*. The function of these proteins is impaired without a small molecule binding to their active site, unlike most of the currently-used β-lactamase inhibitors which often generate resistance (*Laws et al., 2019*). As DsbA dependence is conserved within phylogenetic groups (*Figure 1—figure supplement 2*),

based on the number of enzymes belonging to the same phylogenetic family as the β-lactamases tested in this study (*Supplementary file 1*), we anticipate that a total of 195 discrete enzymes rely on DsbA for their stability and function, 84 of which cannot be inhibited by classical adjuvant approaches. DsbA is widely conserved (*Heras et al., 2009*), thus targeting the DSB system should not only compromise β-lactamases in Enterobacteria but, as demonstrated by our experiments using *P. aeruginosa* clinical isolates (*Figure 8*), could also be a promising avenue for impairing the function of AMR determinants expressed by other highly-resistant Gram-negative organisms. As such, together with the fact that approximately 56% of the β-lactamase phylogenetic families found in pathogens and organisms capable of causing opportunistic infections contain enzymes with two or more cysteines (*Supplementary file 1*), we expect many more clinically relevant β-lactamases, beyond those already tested in this study, to depend on DsbA.

MCR enzymes are rapidly becoming a grave threat to the use of colistin (*Sun et al., 2018*), a drug of last resort often needed for the treatment of multidrug-resistant infections (*Li et al., 2006*). Currently, experimental inhibitors of these proteins are sparse and poorly characterized (*Zhou et al., 2019*), and only one existing compound, the antirheumatic drug auranofin, seems to successfully impair MCR enzymes, through displacement of their zinc cofactor (*Sun et al., 2020*). As all MCR members contain multiple disulfide bonds, inhibition of the DSB system provides a broadly applicable solution for reversing MCR-mediated colistin resistance (*Figures 1C, 5E and 7ABC*) that would likely extend to novel MCR proteins that may emerge in the future. Since the decrease in colistin MIC values upon *dsbA* deletion (*Figure 1C*) or DsbB inhibition (*Figures 5E and 7ABC*) is modest, this phenotype cannot be used in future screens aiming to identify DsbA inhibitors, because such applications require a larger than 4-fold decrease in recorded MIC values to reliably identify promising lead compounds. Nonetheless, our findings in this study clearly demonstrate that absence of DsbA results in degradation of MCR enzymes and abrogation of their function (*Figure 3*), which, in turn, leads to sensitization of all tested *E. coli* clinical isolates to colistin (*Figure 7*). This adds to other efforts aiming to reduce the colistin MIC of polymyxin resistant strains (*Minrovic et al., 2019*; *Zimmerman et al., 2020*). As such, if a clinically useful DsbA inhibitor were to become available, it would be valuable to test its efficacy against large panels of MCR-expressing clinical strains, as it might offer a new way to bypass MCR-mediated colistin resistance.

No clinically applicable efflux pump inhibitors have been identified to date (*Sharma et al., 2019*) despite many efforts to target these macromolecular assemblies as a way to overcome intrinsic resistance. While deletion of *dsbA* sensitizes the tested *E. coli* strain to chloramphenicol, the overall effects of DsbA absence on efflux function are modest at best (*Figure 1D*). That said, our investigation of the relationship between DsbA-mediated proteostasis and pump function (*Figure 4A–C*) highlights the importance of other cell envelope proteins responsible for protein homeostasis, such as DegP, for bacterial efflux. Since the cell envelope contains multiple protein folding catalysts (*Goemans et al., 2014*), it would be worth testing if other redox proteins, chaperones, or proteases could be targeted to indirectly compromise efflux pumps.

More generally, our findings demonstrate that cell envelope proteostasis pathways have significant, yet untapped, potential for the development of novel antibacterial strategies. The example of the DSB system presented here is particularly telling. This pathway, initially considered merely a housekeeping system (*Kadokura et al., 2003*), plays a major role in clinically relevant bacterial niche adaptation. In addition to assisting the folding of 40% of the cell-envelope proteome (*Dutton et al., 2008*; *Vertommen et al., 2008*), the DSB system is essential for virulence (*Heras et al., 2009*; *Landeta et al., 2018*), has a key role in the formation and awakening of bacterial persister cells (*Wilmaerts et al., 2019*) and, as seen in this work, is required for bacterial survival in the presence of widely used antibiotic compounds. As shown in our in vivo experiments (*Figure 8C*), targeting such a system in Gram-negative pathogens could lead to adjuvant approaches that inactivate AMR determinants whilst simultaneously incapacitating an arsenal of virulence factors. Therefore, this study not only lays the groundwork for future clinical applications, such as the development of broad-acting antibiotic adjuvants, but also serves as a paradigm for exploiting other accessible cell envelope proteostasis processes for the design of next-generation therapeutics.

# Materials and methods

**Key resources table**

| Reagent type (species) or resource | Designation | Source or reference | Identifiers | Additional information |
|---|---|---|---|---|
| Genetic reagent (*Escherichia coli*) | DH5α | *Hanahan and Glover, 1985* | F⁻ *endA1 glnV44 thi-1 recA1 relA1 gyrA96 deoR nupG purB20* φ80d*lacZ*ΔM15 Δ(*lacZYA-argF*)U169 *hsd*R17($r_K^-m_K^+$) λ⁻ | - |
| Genetic reagent (*Escherichia coli*) | CC118 λ pir | *Herrero et al., 1990* | *araD* Δ(*ara, leu*) Δ*lacZ74 phoA20 galK thi-1 rspE rpoB argE recA1* λ pir | - |
| Genetic reagent (*Escherichia coli*) | HB101 | *Boyer and Roulland-Dussoix, 1969* | *supE44 hsdS20 recA13 ara-14 proA2 lacY1 galK2 rpsL20 xyl-5 mtl-1* | - |
| Genetic reagent (*Escherichia coli*) | MC1000 | *Casadaban and Cohen, 1980* | *araD139* Δ(*ara, leu*)7697 Δ*lacX74 galU galK strA* | - |
| Genetic reagent (*Escherichia coli*) | MC1000 *dsbA* | *Kadokura et al., 2004* | *dsbA::aphA*, Kan^R | - |
| Genetic reagent (*Escherichia coli*) | MC1000 *dsbA att*Tn7::P*tac-dsbA* | This study | *dsbA::aphA att*Tn7::*dsbA*, Kan^R | Can be obtained from the Mavridou lab |
| Genetic reagent (*Escherichia coli*) | MG1655 | *Blattner et al., 1997* | K-12 F⁻ λ⁻ *ilvG⁻ rfb-50 rph-1* | - |
| Genetic reagent (*Escherichia coli*) | MG1655 *dsbA* | This study | *dsbA::aphA*, Kan^R | Can be obtained from the Mavridou lab |
| Genetic reagent (*Escherichia coli*) | MG1655 *dsbA att*Tn7::P*tac-dsbA* | This study | *dsbA::aphA att*Tn7::*dsbA*, Kan^R | Can be obtained from the Mavridou lab |
| Genetic reagent (*Escherichia coli*) | MG1655 *acrA* | This study | *acrA* | Can be obtained from the Mavridou lab |
| Genetic reagent (*Escherichia coli*) | MG1655 *tolC* | This study | *tolC* | Can be obtained from the Mavridou lab |
| Genetic reagent (*Escherichia coli*) | MG1655 *degP* | This study | *degP::strAB*, Str^R | Can be obtained from the Mavridou lab |
| Strain, strain background (*Escherichia coli*) | BM16 | *Dortet et al., 2014* | *bla*TEM-1b*bla*KPC-2 | Human clinical strain |
| Strain, strain background (*Escherichia coli*) | LIL-1 | *Dortet et al., 2014* | *bla*TEM-1*bla*OXA-9 *bla*KPC-2 | Human clinical strain |
| Strain, strain background (*Escherichia coli*) | CNR1790 | *Dortet et al., 2018* | *bla*TEM-15 *mcr-1* | Human clinical strain |
| Strain, strain background (*Escherichia coli*) | CNR20140385 | *Dortet et al., 2018* | *bla*OXA-48*mcr-1* | Human clinical strain |
| Strain, strain background (*Escherichia coli*) | WI2 (ST1288) | *Beyrouthy et al., 2017* | *bla*OXA-48*bla*KPC-28 *mcr-1* | Human clinical strain |
| Strain, strain background (*Escherichia coli*) | 1073944 (ST117) | *Wise et al., 2018* | *mcr-1* | Human clinical strain |

*Continued on next page*

*Continued*

| Reagent type (species) or resource | Designation | Source or reference | Identifiers | Additional information |
|---|---|---|---|---|
| Strain, strain background (*Escherichia coli*) | 41,489 | **Dortet et al., 2018** | *mcr-1* | Human clinical strain |
| Strain, strain background (*Escherichia coli*) | - | **Dortet et al., 2018** | *mcr-1* | Human clinical strain |
| Strain, strain background (*Escherichia coli*) | 1256822 (ST48) | **Wise et al., 2018** | *mcr-1.5* | Human clinical strain |
| Strain, strain background (*Escherichia coli*) | 27,841 (ST744) | **Haenni et al., 2018** | $bla_{CTX-M-55}mcr-3.2$ | Environmental strain from livestock |
| Strain, strain background (*Escherichia coli*) | 1144230 (ST641) | **Wise et al., 2018** | $bla_{CMY-2}mcr-5$ | Human clinical strain |
| Strain, strain background (*Klebsiella pneumoniae*) | ST234 | **Nordmann et al., 2012** | $bla_{SHV-27}bla_{KPC-2}$ | Human clinical strain |
| Strain, strain background (*Citrobacter freundii*) | BM19 | **Dortet et al., 2014** | $bla_{KPC-2}$ | Human clinical strain |
| Strain, strain background (*Enterobacter cloacae*) | DUB | **Dortet et al., 2015** | $bla_{FRI-1}$ | Human clinical strain |
| Strain, strain background (*Pseudomonas aeruginosa*) | PA43417 | **El Garch et al., 2011** | $bla_{OXA-198}$ | Human clinical strain |
| Genetic reagent (*Pseudomonas aeruginosa*) | PA43417 | This study | $dsbA1\ bla_{OXA-198}$ | Can be obtained from the Mavridou lab |
| Strain, strain background (*Pseudomonas aeruginosa*) | PAe191 | **Mugnier et al., 1998** | $bla_{OXA-19}$ | Human clinical strain |
| Genetic reagent (*Pseudomonas aeruginosa*) | PAe191 | This study | $dsbA1\ bla_{OXA-19}$ | Can be obtained from the Mavridou lab |
| Recombinant DNA reagent | pDM1 (plasmid) | Lab stock | GenBank MN128719 | pDM1 vector, p15A *ori*, P*tac* promoter, MCS, Tet$^R$ |
| Recombinant DNA reagent | pDM1-$bla_{L2-1}$ (plasmid) | This study | - | $bla_{L2-1}$ cloned into pDM1, Tet$^R$; can be obtained from the Mavridou lab |
| Recombinant DNA reagent | pDM1-$bla_{GES-1}$ (plasmid) | This study | - | $bla_{GES-1}$ cloned into pDM1, Tet$^R$; can be obtained from the Mavridou lab |
| Recombinant DNA reagent | pDM1-$bla_{GES-2}$ (plasmid) | This study | - | $bla_{GES-2}$ cloned into pDM1, Tet$^R$; can be obtained from the Mavridou lab |
| Recombinant DNA reagent | pDM1-$bla_{GES-11}$ (plasmid) | This study | - | $bla_{GES-11}$ cloned into pDM1, Tet$^R$; can be obtained from the Mavridou lab |
| Recombinant DNA reagent | pDM1-$bla_{SHV-27}$ (plasmid) | This study | - | $bla_{SHV-27}$ cloned into pDM1, Tet$^R$; can be obtained from the Mavridou lab |

*Continued*

| Reagent type (species) or resource | Designation | Source or reference | Identifiers | Additional information |
|---|---|---|---|---|
| Recombinant DNA reagent | pDM1-$bla_{OXA-4}$ (plasmid) | This study | – | $bla_{OXA-4}$ cloned into pDM1, Tet$^R$; can be obtained from the Mavridou lab |
| Recombinant DNA reagent | pDM1-$bla_{OXA-10}$ (plasmid) | This study | – | $bla_{OXA-10}$ cloned into pDM1, Tet$^R$; can be obtained from the Mavridou lab |
| Recombinant DNA reagent | pDM1-$bla_{OXA-198}$ (plasmid) | This study | – | $bla_{OXA-198}$ cloned into pDM1, Tet$^R$; can be obtained from the Mavridou lab |
| Recombinant DNA reagent | pDM1-$bla_{FRI-1}$ (plasmid) | This study | – | $bla_{FRI-1}$ cloned into pDM1, Tet$^R$; can be obtained from the Mavridou lab |
| Recombinant DNA reagent | pDM1-$bla_{L1-1}$ (plasmid) | This study | – | $bla_{L1-1}$ cloned into pDM1, Tet$^R$; can be obtained from the Mavridou lab |
| Recombinant DNA reagent | pDM1-$bla_{KPC-2}$ (plasmid) | This study | – | $bla_{KPC-2}$ cloned into pDM1, Tet$^R$; can be obtained from the Mavridou lab |
| Recombinant DNA reagent | pDM1-$bla_{KPC-3}$ (plasmid) | This study | – | $bla_{KPC-3}$ cloned into pDM1, Tet$^R$; can be obtained from the Mavridou lab |
| Recombinant DNA reagent | pDM1-$bla_{SME-1}$ (plasmid) | This study | – | $bla_{SME-1}$ cloned into pDM1, Tet$^R$; can be obtained from the Mavridou lab |
| Recombinant DNA reagent | pDM1-*mcr-1* (plasmid) | This study | – | *mcr-1* cloned into pDM1, Tet$^R$; can be obtained from the Mavridou lab |
| Recombinant DNA reagent | pDM1-*mcr-3* (plasmid) | This study | – | *mcr-3* cloned into pDM1, Tet$^R$; can be obtained from the Mavridou lab |
| Recombinant DNA reagent | pDM1-*mcr-4* (plasmid) | This study | – | *mcr-4* cloned into pDM1, Tet$^R$; can be obtained from the Mavridou lab |
| Recombinant DNA reagent | pDM1-*mcr-5* (plasmid) | This study | – | *mcr-5* cloned into pDM1, Tet$^R$; can be obtained from the Mavridou lab |
| Recombinant DNA reagent | pDM1-*mcr-8* (plasmid) | This study | – | *mcr-8* cloned into pDM1, Tet$^R$; can be obtained from the Mavridou lab |
| Recombinant DNA reagent | pDM1-$bla_{L2-1}$-StrepII (plasmid) | This study | – | $bla_{L2-1}$ encoding L2-1 with a C-terminal StrepII tag cloned into pDM1, Tet$^R$; can be obtained from the Mavridou lab |
| Recombinant DNA reagent | pDM1-$bla_{GES-1}$-StrepII (plasmid) | This study | – | $bla_{GES-1}$ encoding GES-1 with a C-terminal StrepII tag cloned into pDM1, Tet$^R$; can be obtained from the Mavridou lab |
| Recombinant DNA reagent | pDM1-StrepII-$bla_{OXA-4}$ (plasmid) | This study | – | $bla_{OXA-4}$ encoding OXA-4 with an N-terminal StrepII tag cloned into pDM1, Tet$^R$; can be obtained from the Mavridou lab |
| Recombinant DNA reagent | pDM1-$bla_{OXA-10}$-StrepII (plasmid) | This study | – | $bla_{OXA-10}$ encoding OXA-10 with a C-terminal StrepII tag cloned into pDM1, Tet$^R$; can be obtained from the Mavridou lab |
| Recombinant DNA reagent | pDM1-$bla_{OXA-198}$-StrepII (plasmid) | This study | – | $bla_{OXA-198}$ encoding OXA-198 with a C-terminal StrepII tag cloned into pDM1, Tet$^R$; can be obtained from the Mavridou lab |
| Recombinant DNA reagent | pDM1-$bla_{FRI-1}$-StrepII (plasmid) | This study | – | $bla_{FRI-1}$ encoding FRI-1 with a C-terminal StrepII tag cloned into pDM1, Tet$^R$; can be obtained from the Mavridou lab |

*Continued*

| Reagent type (species) or resource | Designation | Source or reference | Identifiers | Additional information |
|---|---|---|---|---|
| Recombinant DNA reagent | pDM1-*bla*L1-1-StrepII (plasmid) | This study | - | *bla*L1-1 encoding L1-1 with a C-terminal StrepII tag cloned into pDM1, Tet$^R$; can be obtained from the Mavridou lab |
| Recombinant DNA reagent | pDM1-*bla*KPC-3-StrepII (plasmid) | This study | - | *bla*KPC-3 encoding KPC-3 with a C-terminal StrepII tag cloned into pDM1, Tet$^R$; can be obtained from the Mavridou lab |
| Recombinant DNA reagent | pDM1-*mcr-1*-StrepII (plasmid) | This study | - | *bla*MCR-1 encoding MCR-1 with a C-terminal StrepII tag cloned into pDM1, Tet$^R$; can be obtained from the Mavridou lab |
| Recombinant DNA reagent | pDM1-*mcr-3*-StrepII (plasmid) | This study | - | *bla*MCR-3 encoding MCR-3 with a C-terminal StrepII tag cloned into pDM1, Tet$^R$; can be obtained from the Mavridou lab |
| Recombinant DNA reagent | pDM1-*mcr-4*-StrepII (plasmid) | This study | - | *bla*MCR-4 encoding MCR-4 with a C-terminal StrepII tag cloned into pDM1, Tet$^R$; can be obtained from the Mavridou lab |
| Recombinant DNA reagent | pDM1-*mcr-5*-StrepII (plasmid) | This study | - | *bla*MCR-5 encoding MCR-5 with a C-terminal StrepII tag cloned into pDM1, Tet$^R$; can be obtained from the Mavridou lab |
| Recombinant DNA reagent | pDM1-*mcr-8*-StrepII (plasmid) | This study | - | *bla*MCR-8 encoding MCR-8 with a C-terminal StrepII tag cloned into pDM1, Tet$^R$; can be obtained from the Mavridou lab |
| Recombinant DNA reagent | pGRG25 (plasmid) | **McKenzie and Craig, 2006** | - | Encodes a Tn*7* transposon and *tnsABCD* under the control of P*araB*, thermosensitive pSC101 *ori*, Amp$^R$ |
| Recombinant DNA reagent | pGRG25-P*tac::dsbA* (plasmid) | This study | - | P*tac::dsbA* fragment cloned within the Tn*7* of pGRG25; when inserted into the chromosome and the plasmid cured, the strain expresses DsbA upon IPTG induction, Amp$^R$; can be obtained from the Mavridou lab |
| Recombinant DNA reagent | pSLTS (plasmid) | **Kim et al., 2014** | - | Thermosensitive pSC101*ori*, P*araB* for $\lambda$-Red, P*tetR* for I-SceI, Amp$^R$ |
| Recombinant DNA reagent | pUltraGFP-GM (plasmid) | **Mavridou et al., 2016** | - | Constitutive sfGFP expression from a strong Biofab promoter, p15A *ori*, (template for the *accC* cassette), Gent$^R$ |
| Recombinant DNA reagent | pKD4 (plasmid) | **Datsenko and Wanner, 2000** | - | Conditional oriR$\gamma$ *ori*, (template for the *aphA* cassette), Amp$^R$ |
| Recombinant DNA reagent | pCB112 (plasmid) | **Paradis-Bleau et al., 2014** | | Inducible *lacZ* expression under the control of the P$_{lac}$ promoter, pBR322 *ori*, Cam$^R$ |
| Recombinant DNA reagent | pKNG101 (plasmid) | **Kaniga et al., 1991** | - | Gene replacement suicide vector, *oriR6K*, *oriTRK2*, *sacB*, (template for the *strAB* cassette), Str$^R$ |

*Continued on next page*

*Continued*

| Reagent type (species) or resource | Designation | Source or reference | Identifiers | Additional information |
|---|---|---|---|---|
| Recombinant DNA reagent | pKNG101-*dsbA1* (plasmid) | This study | - | PCR fragment containing the regions upstream and downstream *P. aeruginosa dsbA1* cloned in pKNG101; when inserted into the chromosome the strain is a merodiploid for *dsbA1* mutant, Str$^R$; can be obtained from the Mavridou lab |
| Recombinant DNA reagent | pRK600 (plasmid) | *Kessler et al., 1992* | - | Helper plasmid, ColE1 *ori*, *mob*RK2, *tra*RK2, Cam$^R$ |
| Recombinant DNA reagent | pMA-T *mcr-3* (plasmid) | This study | - | GeneArt cloning vector containing *mcr-3*, ColE1 *ori*, (template for *mcr-3*), Amp$^R$; can be obtained from the Mavridou lab |
| Recombinant DNA reagent | pMK-T *mcr-8* (plasmid) | This study | - | GeneArt cloning vector containing *mcr-8*, ColE1 *ori*, (template for *mcr-8*), Kan$^R$; can be obtained from the Mavridou lab |
| Chemical compound, drug | Ampicillin | Melford | A40040-10.0 | - |
| Chemical compound, drug | Piperacillin | Melford | P55100-1.0 | - |
| Chemical compound, drug | Cefuroxime | Melford | C56300-1.0 | - |
| Chemical compound, drug | Ceftazidime | Melford | C59200-5.0 | - |
| Chemical compound, drug | Imipenem | Cambridge Bioscience | CAY16039-100 mg | - |
| Chemical compound, drug | Aztreonam | Cambridge Bioscience | CAY19784-100 mg | - |
| Chemical compound, drug | Kanamycin | Gibco | 11815032 | - |
| Chemical compound, drug | Gentamicin | VWR | A1492.0025 | - |
| Chemical compound, drug | Streptomycin | ACROS Organics | AC612240500 | - |
| Chemical compound, drug | Tetracycline | Duchefa Biochemie | T0150.0025 | - |
| Chemical compound, drug | Colistin sulphate | Sigma | C4461-1G | - |
| Chemical compound, drug | Tazobactam | Sigma | T2820-10MG | - |
| Chemical compound, drug | Isopropyl β-D-1-thiogalactopyranoside (IPTG) | Melford | I56000-25.0 | - |
| Chemical compound, drug | KOD Hotstart DNA Polymerase | Sigma | 71086–3 | - |
| Chemical compound, drug | Nitrocefin | Abcam | ab145625-25mg | - |
| Chemical compound, drug | 1-N-phenylnaphthylamine (NPN) | Acros Organics | 147160250 | - |

*Continued*

| Reagent type (species) or resource | Designation | Source or reference | Identifiers | Additional information |
|---|---|---|---|---|
| Chemical compound, drug | 4-acetamido-4′-maleimidyl-stilbene-2,2′-disulfonic acid (AMS) | ThermoFisher Scientific | A485 | - |
| Chemical compound, drug | 4,5-dichloro-2-(2-chlorobenzyl) pyridazin-3-one | Enamine | EN300-173996 | - |
| Commercial assay or kit | BugBuster Mastermix | Sigma | 71456–3 | - |
| Commercial assay or kit | Novex ECL HRP chemiluminescent substrate reagent kit | ThermoFisher Scientific | WP20005 | - |
| Commercial assay or kit | SigmaFast BCIP/NBT tablets | Sigma | B5655-25TAB | - |
| Commercial assay or kit | Immobilon Crescendo chemiluminescent reagent | Sigma | WBLUR0100 | - |
| Commercial assay or kit | ETEST - Amoxicillin | Biomerieux | 412,242 | - |
| Commercial assay or kit | ETEST - Cefuroxime | Biomerieux | 412,304 | - |
| Commercial assay or kit | ETEST - Ceftazidime | Biomerieux | 412,292 | - |
| Commercial assay or kit | ETEST - Imipenem | Biomerieux | 412,373 | - |
| Commercial assay or kit | ETEST - Aztreonam | Biomerieux | 412,258 | - |
| Commercial assay or kit | ETEST - Gentamicin | Biomerieux | 412,367 | - |
| Commercial assay or kit | ETEST - Erythromycin | Biomerieux | 412,333 | - |
| Commercial assay or kit | ETEST - Chloramphenicol | Biomerieux | 412,308 | - |
| Commercial assay or kit | ETEST - Nalidixic acid | Biomerieux | 516,540 | - |
| Commercial assay or kit | ETEST - Ciprofloxacin | Biomerieux | 412,310 | - |
| Commercial assay or kit | ETEST - Nitrofurantoin | Biomerieux | 530,440 | - |
| Commercial assay or kit | ETEST - Trimethoprim | Biomerieux | 412,482 | - |
| Antibody | Strep-Tactin-HRP conjugate (mouse monoclonal) | Iba Lifesciences | NC9523094 | (1:3,000) in 3 w/v % BSA/TBS-T |
| Antibody | Strep-Tactin-AP conjugate (mouse monoclonal) | Iba Lifesciences | NC0485490 | (1:3,000) in 3 w/v % BSA/TBS-T |
| Antibody | anti-DsbA (rabbit polyclonal) | Beckwith lab | - | (1:1,000) in 5 w/v % skimmed milk/TBS-T |
| Antibody | anti-AcrA (rabbit polyclonal) | Koronakis lab | - | (1:10,000) in 5 w/v % skimmed milk/TBS-T |
| Antibody | anti-TolC (rabbit polyclonal) | Koronakis lab | - | (1:5,000) in 5 w/v % skimmed milk/TBS-T |
| Antibody | anti-HtrA1 (DegP) (rabbit polyclonal) | Abcam | ab231195 | (1:1,000) in 5 w/v % skimmed milk/TBS-T |
| Antibody | anti-DnaK 8E2/2 (mouse monoclonal) | Enzo Life Sciences | ADI-SPA-880-D | (1:10,000) in 5% w/v skimmed milk/TBS-T |
| Antibody | anti-rabbit IgG-AP conjugate (goat polyclonal) | Sigma | A3687-.25ML | (1:6,000) in 5% w/v skimmed milk/TBS-T |
| Antibody | anti-rabbit IgG-HRP conjugate (goat polyclonal) | Sigma | A0545-1ML | (1:6,000) in 5% w/v skimmed milk/TBS-T |
| Antibody | anti-mouse IgG-AP conjugate (goat polyclonal) | Sigma | A3688-.25ML | (1:6,000) in 5% w/v skimmed milk/TBS-T |

*Continued on next page*

*Continued*

| Reagent type (species) or resource | Designation | Source or reference | Identifiers | Additional information |
|---|---|---|---|---|
| Antibody | anti-mouse IgG-HRP conjugate (goat polyclonal) | Sigma | A4416-.5ML | (1:6,000) in 5% w/v skimmed milk/ TBS-T |
| Software, algorithm | FlowJo | Tree Star | - | version 10.0.6 |
| Software, algorithm | Adobe Photoshop CS4 | Adobe | - | extended version 11.0 |
| Software, algorithm | Prism | GraphPad | - | version 8.0.2 |
| Software, algorithm | blastp | *Altschul et al., 1990* | - | version 2.2.28+ |
| Software, algorithm | USEARCH | *Edgar, 2010* | - | version 7.0 |
| Software, algorithm | MUSCLE | *Edgar, 2004* | - | - |
| Software, algorithm | FastTree | *Price et al., 2010* | - | version 2.1.7 |
| Software, algorithm | HMMER | *Finn et al., 2015* | - | version 3.1b2 |

## Reagents and bacterial growth conditions

Unless otherwise stated, chemicals and reagents were acquired from Sigma Aldrich, growth media were purchased from Oxoid and antibiotics were obtained from Melford Laboratories. Lysogeny broth (LB) (10 g/L NaCl) and agar (1.5% w/v) were used for routine growth of all organisms at 37 °C with shaking at 220 RPM, as appropriate. Unless otherwise stated, Mueller-Hinton (MH) broth and agar (1.5% w/v) were used for Minimum Inhibitory Concentration (MIC) assays. Growth media were supplemented with the following, as required: 0.25 mM Isopropyl β-D-1-thiogalactopyranoside (IPTG) (for strains harboring β-lactamase-encoding pDM1 plasmids), 0.5 mM IPTG (for strains harboring MCR-encoding pDM1 plasmids), 12.5 μg/mL tetracycline, 100 μg/mL ampicillin, 50 μg/mL kanamycin, 10 μg/mL gentamicin, 33 μg/mL chloramphenicol, 50 μg/mL streptomycin (for cloning purposes), and 2000–5000 μg/mL streptomycin (for the construction of *Pseudomonas aeruginosa* mutants).

## Construction of plasmids and bacterial strains

Bacterial strains and plasmids used in this study are listed in the Key Resources Table and in *Supplementary file 3* - Supplementary Tables 2 and 3, respectively. Oligonucleotides used in this study are listed in Supplementary Table 4. DNA manipulations were conducted using standard methods. KOD Hot Start DNA polymerase (Merck) was used for all PCR reactions according to the manufacturer's instructions, oligonucleotides were synthesized by Sigma Aldrich and restriction enzymes were purchased from New England Biolabs. All DNA constructs were sequenced and confirmed to be correct before use.

Genes for β-lactamase and MCR enzymes were amplified from genomic DNA extracted from clinical isolates (*Supplementary file 3* - Supplementary Table 5) with the exception of *mcr-3* and *mcr-8*, which were synthesized by GeneArt Gene Synthesis (ThermoFisher Scientific). β-lactamase and MCR genes were cloned into the IPTG-inducible plasmid pDM1 using primers P1-P34. pDM1 (GenBank accession number MN128719) was constructed from the p15A-*ori* plasmid pACYC184 (*Chang and Cohen, 1978*) to contain the Lac repressor, the P*tac* promoter, an optimized ribosome binding site and a multiple cloning site (NdeI, SacI, PstI, KpnI, XhoI, and XmaI) inserted into the NcoI restriction site of pACYC184. All StrepII-tag fusions of β-lactamase and MCR enzymes (constructed using primers P1, P3, P9, P11, P13, P15, P17, P21, P23, P25, P27, P29, P35, P36, and P39-P48) have a C-terminal StrepII tag (GSAWSHPQFEK) except for OXA-4, where an N-terminal StrepII tag was inserted between the periplasmic signal sequence and the body of the protein using the primer pairs P7/P38, P9/P37, and P7/P8. Plasmids encoding *ges-1* and *kpc-3* were obtained by performing QuickChange mutagenesis on pDM1 constructs encoding *ges-5* and *kpc-2*, respectively (primers P31-P34).

*E. coli* gene mutants were constructed using a modified lambda-Red recombination method, as previously described (*Kim et al., 2014*) (primers P51-P58). To complement the *dsbA* mutant, a DNA fragment consisting of *dsbA* preceded by the P*tac* promoter was inserted into the NotI/XhoI sites of pGRG25 (primers P49/P50) and was reintroduced into the *E. coli* chromosome at the *att*Tn7 site, as previously described (*McKenzie and Craig, 2006*). The *dsbA1* mutants of the *P. aeruginosa*

PA43417 and *P. aeruginosa* PAe191 clinical isolates were constructed by allelic exchange, as previously described (*Vasseur et al., 2005*). Briefly, the *dsbA1* gene area of *P. aeruginosa* PA43417 and *P. aeruginosa* PAe191 (including the *dsbA1* gene and 600 bp on either side of this gene) was amplified (primers P59/P60) and the obtained DNA was sequenced to allow for accurate primer design for the ensuing cloning step. Subsequently, 500 bp DNA fragments upstream and downstream of the *dsbA1* gene were amplified using *P. aeruginosa* PA43417 genomic DNA (primers P61/P62 [upstream] and P63/P64 [downstream]). A fragment containing both regions was obtained by overlapping PCR (primers P61/P64) and inserted into the XbaI/BamHI sites of pKNG101. The suicide vector pKNG101 (*Kaniga et al., 1991*) is not replicative in *P. aeruginosa*; it was maintained in *E. coli* CC118 $\lambda$ pir and mobilized into *P. aeruginosa* PA43417 and *P. aeruginosa* PAe191 by triparental conjugation.

## MIC assays

Unless otherwise stated, antibiotic MIC assays were carried out in accordance with the EUCAST recommendations using ETEST strips (BioMérieux). Briefly, overnight cultures of each strain to be tested were standardized to $OD_{600}$ 0.063 in 0.85% NaCl (equivalent to McFarland standard 0.5) and distributed evenly across the surface of MH agar plates. ETEST strips were placed on the surface of the plates, evenly spaced, and the plates were incubated for 18–24 hr at 37 °C. MICs were read according to the manufacturer's instructions. β-lactam MICs were also determined using the Broth Microdilution (BMD) method, as required. Briefly, a series of antibiotic concentrations was prepared by twofold serial dilution in MH broth in a clear-bottomed 96-well microtiter plate (Corning). When used, tazobactam was included at a fixed concentration of 4 µg/mL in every well, in accordance with the EUCAST guidelines. The strain to be tested was added to the wells at approximately $5 \times 10^4$ colony-forming units (CFU) per well and plates were incubated for 18–24 hr at 37 °C. The MIC was defined as the lowest antibiotic concentration with no visible bacterial growth in the wells. Vancomycin MICs were determined using the BMD method, as above. All colistin sulphate MIC assays were performed using the BMD method as described above except that instead of twofold serial dilutions, the following concentrations of colistin (Acros Organics) were prepared individually in MH broth: 32 µg/mL, 16 µg/mL, 12 µg/mL, 8 µg/mL, 7 µg/mL, 6 µg/mL, 5.5 µg/mL, 5 µg/mL, 4.5 µg/mL, 4 µg/mL, 3.5 µg/mL, 3 µg/mL, 2.5 µg/mL, 2 µg/mL, 1.5 µg/mL, 1 µg/mL, 0.5 µg/mL.

The covalent DsbB inhibitor 4,5-dichloro-2-(2-chlorobenzyl)pyridazin-3-one (*Landeta et al., 2015*) was used to chemically impair the function of the DSB system. Inactivation of DsbB results in abrogation of DsbA function (*Kishigami et al., 1995*) only in media free of small-molecule oxidants (*Dailey and Berg, 1993*). Therefore, MIC assays involving chemical inhibition of the DSB system were performed using M63 broth (15.1 mM $(NH_4)_2SO_4$, 100 mM $KH_2PO_4$, 1.8 mM $FeSO_4.7H_2O$, adjusted to pH 7.2 with KOH) and agar (1.5% w/v) supplemented with 1 mM $MgSO_4$, 0.02% w/v glucose, 0.005% w/v thiamine, 31 µM $FeCl_3.6H_2O$, 6.2 µM $ZnCl_2$, 0.76 µM $CuCl_2.2H_2O$, 1.62 µM $H_3BO_3$, 0.081 µM $MnCl_2.4H_2O$, 84.5 mg/L alanine, 19.5 mg/L arginine, 91 mg/L aspartic acid, 65 mg/L glutamic acid, 78 mg/L glycine, 6.5 mg/L histidine, 26 mg/L isoleucine, 52 mg/L leucine, 56.34 mg/L lysine, 19.5 mg/L methionine, 26 mg/L phenylalanine, 26 mg/L proline, 26 mg/L serine, 6.5 mg/L threonine, 19.5 mg/L tyrosine, 56.34 mg/L valine, 26 mg/L tryptophan, 26 mg/L asparagine and 26 mg/L glutamine. $CaCl_2$ was also added at a final concentration of 0.223 mM for colistin sulfate MIC assays. Either DMSO (vehicle control) or the covalent DsbB inhibitor 4,5-dichloro-2-(2-chlorobenzyl)pyridazin-3-one (final concentration of 50 µM) (Enamine) (*Landeta et al., 2015*) were added to the M63 medium, as required. The strain to be tested was added at an inoculum that recapitulated the MH medium MIC values obtained for that strain.

## SDS-PAGE analysis and immunoblotting

Samples for immunoblotting were prepared as follows. Strains to be tested were grown on LB or MH agar plates as lawns in the same manner as for MIC assays described above. Bacteria were collected using an inoculating loop and resuspended in 0.85% NaCl or LB to $OD_{600}$ 2.0 (except for strains expressing OXA-4, where $OD_{600}$ 6.0 was used). For strains expressing β-lactamase enzymes, the cell suspensions were spun at 10,000 *x g* for 10 min and bacterial pellets were lysed by addition of BugBuster Master Mix (Merck Millipore) for 25 min at room temperature with gentle agitation. Subsequently, lysates were spun at 10,000 *x g* for 10 min at 4 °C and the supernatant was added to 4 x Laemmli buffer. For strains expressing MCR enzymes cell suspensions were directly added to 4 x

Laemmli buffer, while for *E. coli* MG1655 and its mutants, cells were lysed as above and lysates were added to 4 x Laemmli buffer. All samples were boiled for 5 min before separation by SDS-PAGE.

Unless otherwise stated, SDS-PAGE analysis was carried out using 10% BisTris NuPAGE gels (ThermoFisher Scientific) using MES/SDS running buffer prepared according to the manufacturer's instructions and including pre-stained protein markers (SeeBlue Plus 2, ThermoFisher Scientific). Proteins were transferred to Amersham Protran nitrocellulose membranes (0.45 µm pore size, GE Life Sciences) using a Trans-Blot Turbo transfer system (Bio-Rad) before blocking in 3% w/v Bovine Serum Albumin (BSA)/TBS-T (0.1 % v/v Tween 20) or 5% w/v skimmed milk/TBS-T and addition of primary and secondary antibodies. The following primary antibodies were used in this study: Strep-Tactin-HRP conjugate (Iba Lifesciences) (dilution 1:3,000 in 3 w/v % BSA/TBS-T), Strep-Tactin-AP conjugate (Iba Lifesciences) (dilution 1:3,000 in 3 w/v % BSA/TBS-T), rabbit anti-DsbA antibody (dilution 1:1,000 in 5 w/v % skimmed milk/TBS-T), rabbit anti-AcrA antibody (dilution 1:10,000 in 5 w/v % skimmed milk/TBS-T), rabbit anti-TolC antibody (dilution 1:5,000 in 5 w/v % skimmed milk/TBS-T), rabbit anti-HtrA1 (DegP) antibody (Abcam) (dilution 1:1,000 in 5 w/v % skimmed milk/TBS-T) and mouse anti-DnaK 8E2/2 antibody (Enzo Life Sciences) (dilution 1:10,000 in 5% w/v skimmed milk/TBS-T). The following secondary antibodies were used in this study: goat anti-rabbit IgG-AP conjugate (Sigma Aldrich) (dilution 1:6,000 in 5% w/v skimmed milk/TBS-T), goat anti-rabbit IgG-HRP conjugate (Sigma Aldrich) (dilution 1:6,000 in 5% w/v skimmed milk/TBS-T), goat anti-mouse IgG-AP conjugate (Sigma Aldrich) (dilution 1:6,000 in 5% w/v skimmed milk/TBS-T) and goat anti-mouse IgG-HRP conjugate (Sigma Aldrich) (dilution 1:6000 in 5% w/v skimmed milk/TBS-T). Membranes were washed three times for 5 min with TBS-T prior to development. Development for AP conjugates was carried out using a SigmaFast BCIP/NBT tablet, while HRP conjugates were visualized with the Novex ECL HRP chemiluminescent substrate reagent kit (ThermoFisher Scientific) or the Immobilon Crescendo chemiluminescent reagent (Merck) using a Gel Doc XR + Imager (Bio-Rad).

## β-Lactam hydrolysis assay

β-lactam hydrolysis measurements were carried out using the chromogenic β-lactam nitrocefin (Abcam). Briefly, overnight cultures of strains to be tested were centrifugated, pellets were weighed and resuspended in 150 µL of 100 mM sodium phosphate buffer (pH 7.0) per 1 mg of wet-cell pellet, and cells were lysed by sonication. For strains harboring pDM1, pDM1-$bla_{L2-1}$, pDM1-$bla_{OXA-10}$, and pDM1-$bla_{GES-1}$, lysates corresponding to 0.34 mg of bacterial pellet were transferred into clear-bottomed 96-well microtiter plates (Corning). For strains harboring pDM1-$bla_{OXA-4}$ and pDM1-$bla_{OXA-198}$, lysates corresponding to 0.2 mg and 0.014 mg of bacterial pellet were used, respectively. In all cases, nitrocefin was added at a final concentration of 400 µM and the final reaction volume was made up to 100 µL using 100 mM sodium phosphate buffer (pH 7.0). Nitrocefin hydrolysis was monitored at 25 °C by recording absorbance at 490 nm at 60-s intervals for 15 min using an Infinite M200 Pro microplate reader (Tecan). The amount of nitrocefin hydrolyzed by each lysate in 15 min was calculated using a standard curve generated by acid hydrolysis of nitrocefin standards.

## NPN uptake assay

1-N-phenylnaphthylamine (NPN) (Acros Organics) uptake assays were performed performed as previously described (*Helander and Mattila-Sandholm, 2000*). Briefly, mid-log phase cultures of strains to be tested were diluted to $OD_{600}$ 0.5 in 5 mM HEPES (pH 7.2) before transfer to clear-bottomed 96-well microtiter plates (Corning) and addition of NPN at a final concentration of 10 µM. Colistin sulphate (Acros Organics) was included at a final concentration of 0.5 µg/mL, as required. Immediately after the addition of NPN, fluorescence was measured at 60-s intervals for 10 min using an Infinite M200 Pro microplate reader (Tecan); the excitation wavelength was set to 355 nm and emission was recorded at 405 nm.

## PI uptake assay

Exponentially-growing ($OD_{600}$ 0.4) *E. coli* strains harboring pUltraGFP-GM (*Mavridou et al., 2016*) were diluted to $OD_{600}$ 0.1 in phosphate buffered saline (PBS) (pH 7.4) and cecropin A was added to a final concentration of 20 µM, as required. Cell suspensions were incubated at room temperature for 30 min before centrifugation and resuspension of the pellets in PBS. Propidium iodide (PI) was then added at a final concentration of 3 µM. Suspensions were incubated for 10 min at room temperature

and analyzed on a two-laser, four-color BD FACSCalibur flow cytometer (BD Biosciences). 50,000 events were collected for each sample and data were analyzed using FlowJo v.10.0.6 (Treestar).

## CPRG hydrolysis assay

The cell envelope integrity of bacterial strains used in this study and of their *dsbA* mutants, was tested by measuring the hydrolysis of the β-galactosidase substrate chlorophenyl red-β-D-galactopyranoside (CPRG) by cytoplasmic LacZ, as previously described (*Paradis-Bleau et al., 2014*). Briefly, exponentially growing (OD$_{600}$ 0.4) *E. coli* MC1000 harboring pCB112 or MG1655, as well as their *dsbA* mutants, were diluted to 1:10$^5$ in MH broth and plated on MH agar containing CPRG and IPTG at final concentrations of 20 µg/mL and 50 µM, respectively. Plates were incubated at 37 °C for 18 hr, were photographed, and images were analyzed using Adobe Photoshop CS4 extended v.11.0 (Adobe) as follows. Images were converted to CMYK color space format, colonies were manually selected using consistent tolerance (26, anti-alias, contiguous) and edge refinement (32 px, 100% contrast), and the magenta color was quantified for each image and normalized for the area occupied by each colony.

## MALDI-TOF mass spectrometry

Lipid A profiles of strains to be tested were determined using intact bacteria, as previously described (*Larrouy-Maumus et al., 2016*). The peak for *E. coli* native lipid A is detected at *m/z* 1796.2, whereas the lipid A profiles of strains expressing functional MCR enzymes have two additional peaks, at *m/z* 1821.2 and 1919.2. These peaks result from MCR-mediated modification of native lipid A through addition of phosphoethanolamine moieties (*Dortet et al., 2018*). The ratio of modified to unmodified lipid A was calculated by summing the intensities of the peaks at *m/z* 1821.2 and 1919.2 and dividing this value by the intensity of the native lipid A peak at *m/z* 1796.2.

## Motility Assay

A total of 500 µL of overnight culture of each strain to be tested were centrifuged and the pellets were washed three times in M63 broth before resuspension in the same medium to achieve a final volume of 25 µL. Bacterial motility was assessed by growth in M63 medium containing 0.25% w/v agar supplemented as described above. DMSO (vehicle control) or the covalent DsbB inhibitor 4,5-dichloro-2-(2-chlorobenzyl)pyridazin-3-one (final concentration of 50 µM) (Enamine) were added to the medium, as required. One µL of the washed cell suspension was inoculated into the center of a 90-mm diameter agar plate, just below the surface of the semi-solid medium. Plates were incubated at 37 °C in a humidified environment for 16–18 hr and growth halo diameters were measured.

## AMS labeling

Bacterial strains to be tested were grown for 18 hr in M63 broth supplemented as described above. DMSO (vehicle control) or the covalent DsbB inhibitor 4,5-dichloro-2-(2-chlorobenzyl)pyridazin-3-one (final concentration of 50 µM) (Enamine) were added to the medium, as required. Cultures were standardized to OD$_{600}$ 2.0 in M63 broth, spun at 10,000 *x g* for 10 min and bacterial pellets lysed by addition of BugBuster Master Mix (Merck Millipore) for 25 min at room temperature with gentle agitation. Subsequently, lysates were spun at 10,000 *x g* for 10 min at 4 °C prior to reaction with 4-acetamido-4′-maleimidyl-stilbene-2,2′-disulfonic acid (AMS) (ThermoFisher Scientific). AMS alkylation was performed by vortexing the lysates in 15 mM AMS, 50 mM Tris-HCl, 3% w/v SDS and 3 mM EDTA (pH 8.0) for 30 min at 25 °C, followed by incubation at 37 °C for 10 min. SDS-PAGE analysis and immunoblotting was carried out as described above, except that 12% BisTris NuPAGE gels (ThermoFisher Scientific) and MOPS/SDS running buffer were used. DsbA was detected using a rabbit anti-DsbA primary antibody and an AP-conjugated secondary antibody, as described above.

## Bacterial growth assays

To assess the effect of DSB system inhibition of the growth of *E. coli*, overnight cultures of the strains to be tested were centrifuged and the pellets were washed three times in M63 broth before transfer to clear-bottomed 96-well microtiter plates (Corning) at approximately 5 × 10$^7$ CFU/well (starting OD$_{600}$ ~0.03). M63 broth supplemented as described above was used as a growth medium. DMSO (vehicle control) or the covalent DsbB inhibitor 4,5-dichloro-2-(2-chlorobenzyl)pyridazin-3-one (final concentration of 50 µM) (Enamine) were added to the medium, as required. Plates were incubated

at 37 °C with orbital shaking (amplitude 3 mm, equivalent to ~220 RPM) and $OD_{600}$ was measured at 900-s intervals for 18 hr using an Infinite M200 Pro microplate reader (Tecan). The same experimental setup was also used for recording growth curves of *E. coli* strains and their isogenic mutants, except that overnight cultures of the strains to be tested were diluted 1:100 into clear-bottomed 96-well microtiter plates (Corning) (starting $OD_{600}$ ~0.01) and that LB was used as the growth medium.

### *Galleria mellonella* survival assay

The wax moth model *Galleria mellonella* was used for in vivo survival assays (**McCarthy et al., 2017**). Individual *G. mellonella* larvae were randomly allocated to experimental groups; no masking was used. Overnight cultures of the strains to be tested were standardized to $OD_{600}$ 1.0, suspensions were centrifuged and the pellets were washed three times in PBS and serially diluted. Ten µl of the 1:10 dilution of each bacterial suspension was injected into the last right abdominal proleg of 30 *G. mellonella* larvae per condition; an additional 10 larvae were injected with PBS as negative control. Immediately after infection, larvae were injected with 4 µl of ceftazidime to a final concentration of 7.5 µg/ml in the last left abdominal proleg. The larvae mortality was monitored for 50 hr. Death was scored when larvae turned black due to melanization, and did not respond to physical stimulation.

### SEM imaging

Bacterial strains to be tested were grown for 18 hr in MH broth; the covalent DsbB inhibitor 4,5-dichloro-2-(2-chlorobenzyl)pyridazin-3-one (final concentration of 50 µM) (Enamine) was added to the medium, as required. Cells were centrifuged, the pellets were washed three times in M63 broth, and cell suspensions were diluted 1:500 into the same medium supplemented as described above; the covalent DsbB inhibitor (final concentration of 50 µM) and/or antibiotics (final concentrations of 6 µg/mL and 2 µg/mL of imipenem and colistin, respectively) were added to the cultures, as required. After 1 hr of incubation as described above, 25 µl of each culture was spotted onto positively charged glass microscope slides and allowed to air-dry. Cells were then fixed with glutaraldehyde (2.5% v/v in PBS) for 30 min at room temperature and the slide was washed five times in PBS. Subsequently, each sample was dehydrated using increasing concentrations of ethanol (5% v/v, 10% v/v, 20% v/v, 30% v/v, 50% v/v, 70% v/v, 90% v/v [applied three times] and 100% v/v), with each wash being carried out by application and immediate removal of the washing solution, before a 7-nm coat of platinum/palladium was applied using a Cressington 208 benchtop sputter coater. Images were obtained on a Zeiss Supra 40 V Scanning Electron Microscope at 5.00 kV and with ×26,000 magnification.

### Statistical analysis of experimental data

The total numbers of performed biological experiments and technical repeats are mentioned in the figure legend of each display item. Biological replication refers to completely independent repetition of an experiment using different biological and chemical materials. Technical replication refers to independent data recordings using the same biological sample. For MIC assays, all recorded values are displayed in the relevant graphs; for MIC assays where three or more biological experiments were performed, the bars indicate the median value, while for assays where two biological experiments were performed the bars indicate the most conservative of the two values (i.e. for increasing trends, the value representing the smallest increase and for decreasing trends, the value representing the smallest decrease). For all other assays, statistical analysis was performed in GraphPad Prism v8.0.2 using an unpaired T-test with Welch's correction, a one-way ANOVA with correction for multiple comparisons, or a Mantel-Cox logrank test, as appropriate. Statistical significance was defined as $p < 0.05$. Outliers were defined as any technical repeat >2 SD away from the average of the other technical repeats within the same biological experiment. Such data were excluded and all remaining data were included in the analysis. Detailed information for each figure is provided below:

*Figure 2C* unpaired T-test with Welch's correction; n = 3; 3.621 degrees of freedom, t-value = 0.302, p = 0.7792 (non-significance) (for pDM1 strains); 3.735 degrees of freedom, t-value = 0.4677, p = 0.666 (non-significance) (for pDM1-$bla_{L2-1}$ strains); 2.273 degrees of freedom, t-value = 5.069, p = 0.0281 (significance) (for pDM1-$bla_{GES-1}$ strains); 2.011 degrees of freedom, t-value = 6.825, p = 0.0205 (significance) (for pDM1-$bla_{OXA-4}$ strains); 2.005 degrees of freedom, t-value = 6.811, p = 0.0208 (significance) (for pDM1-$bla_{OXA-10}$ strains); 2.025 degrees of freedom, t-value = 5.629, p = 0.0293 (significance) (for pDM1-$bla_{OXA-198}$ strains).

*Figure 3C* one-way ANOVA with Tukey's multiple comparison test; n = 4; 24 degrees of freedom; F value = 21.00; p = 0.000000000066 (for pDM1-*mcr-3* strains), p = 0.0004 (for pDM1-*mcr-4* strains), p = 0.000000000066 (for pDM1-*mcr-5* strains), p = 0.00066 (for pDM1-*mcr-8* strains).

*Figure 5B* one-way ANOVA with Bonferroni's multiple comparison test; n = 3; 6 degrees of freedom; F value = 1878; p = 0.000000002 (significance).

*Figure 8C* Mantel-Cox test; n = 30; p =< 0.0001 (significance) (*P. aeruginosa* versus *P. aeruginosa dsbA1*), p > 0.9999 (non-significance) (*P. aeruginosa* vs *P. aeruginosa* treated with ceftazidime), p =< 0.0001 (significance) (*P. aeruginosa* treated with ceftazidime versus *P. aeruginosa dsbA1*), p =< 0.0001 (significance) (*P. aeruginosa dsbA1* versus *P. aeruginosa dsbA1* treated with ceftazidime).

*Figure 1—figure supplement 7A* (left graph): one-way ANOVA with Bonferroni's multiple comparison test; n = 3; 6 degrees of freedom; F value = 39.22; p = 0.0007 (significance), p = 0.99 (non-significance).

*Figure 1—figure supplement 7B* (left graph): one-way ANOVA with Bonferroni's multiple comparison test; n = 3; 6 degrees of freedom; F value = 61.84; p = 0.0002 (significance), p = 0.99 (non-significance).

*Figure 1—figure supplement 7B* (right graph): unpaired T-test with Welch's correction, n = 3; 4 degrees of freedom; t-value = 0.1136, p = 0.9150 (non-significance).

*Figure 1—figure supplement 9A* (left graph): one-way ANOVA with Bonferroni's multiple comparison test; n = 3; 6 degrees of freedom; F value = 261.4; p = 0.00000055 (significance), p = 0.0639 (non-significance).

*Figure 1—figure supplement 9B* (left graph): one-way ANOVA with Bonferroni's multiple comparison test; n = 3; 6 degrees of freedom; F value = 77.49; p = 0.0001 (significance), p = 0.9999 (non-significance).

*Figure 1—figure supplement 9B* (right graph): unpaired T-test with Welch's correction, n = 3; 4 degrees of freedom; t-value = 0.02647, p = 0.9801 (non-significance).

## Bioinformatics

The following bioinformatics analyses were performed in this study. Short scripts and pipelines were written in Perl (version 5.18.2) and executed on macOS Sierra 10.12.5.

*β-lactamase enzymes.* All available protein sequences of β-lactamases were downloaded from http://www.bldb.eu (*Naas et al., 2017*) (5 August 2021). Sequences were clustered using the ucluster software with a 90% identity threshold and the cluster_fast option (USEARCH v.7.0 (77)); the centroid of each cluster was used as a cluster identifier for every sequence. All sequences were searched for the presence of cysteine residues using a Perl script. Proteins with two or more cysteines after the first 30 amino acids of their primary sequence were considered potential substrates of the DSB system for organisms where oxidative protein folding is carried out by DsbA and provided that translocation of the β-lactamase outside the cytoplasm is performed by the Sec system. The first 30 amino acids of each sequence were excluded to avoid considering cysteines that are part of the signal sequence mediating the translocation of these enzymes outside the cytoplasm. The results of the analysis can be found in *Supplementary file 1*.

*MCR enzymes.* *E. coli* MCR-1 (AKF16168.1) was used as a query in a blastp 2.2.28+ (*Altschul et al., 1990*) search limited to *Proteobacteria* on the NCBI Reference Sequence (RefSeq) proteome database (21 April 2019) (evalue <10e-5). A total of 17,503 hit sequences were retrieved and clustered using the ucluster software with a 70% identity threshold and the cluster_fast option (USEARCH v.7.0 (77)). All centroid sequences were retrieved and clustered again with a 20% identity threshold and the cluster_fast option. Centroid sequences of all clusters comprising more than five sequences (809 sequences retrieved) along with the sequences of the five MCR enzymes tested in this study were aligned using MUSCLE (*Edgar, 2004*). Sequences which were obviously divergent or truncated were manually eliminated and a phylogenetic tree was built from a final alignment comprising 781 sequences using FastTree 2.1.7 with the wag substitution matrix and default parameters (*Price et al., 2010*). The assignment of each protein sequence to a specific group was done using hmmsearch (HMMER v.3.1b2) (*Finn et al., 2015*) with Hidden Markov Models built from confirmed sequences of MCR-like and EptA-like proteins.

## Acknowledgements

We thank J Rowley for assistance with flow cytometry. SEM imaging was performed at the University of Texas Center for Biomedical Research Support core; we thank M Mikesh for providing the training for the acquisition of SEM images. We are grateful to IHMA Inc Schaumburg for the kind gift of the *E. coli* 1144230 isolate, T Bernhardt for the kind gift of the pCB112 plasmid, and J Beckwith, F Alcock and V Koronakis for the kind gifts of the anti-DsbA, the anti-AcrA and the anti-TolC antibodies, respectively.

Research reported in this publication was supported by the National Institute of Allergy and Infectious Diseases of the National Institutes of Health under Award Number R01AI158753 (to DAIM). The content is solely the responsibility of the authors and does not necessarily represent the official views of the National Institutes of Health. This study was also funded by the MRC Career Development Award MR/M009505/1 (to DAIM), the institutional BBSRC-DTP studentships BB/M011178/1 (to NK) and BB/M01116X/1 (to HLP), the BBSRC David Philips Fellowship BB/M02623X/1 (to JMAB), the ISSF Wellcome Trust grant 105603/Z/14/Z (to GL-M), the British Society for Antimicrobial Chemotherapy, NC3Rs, BBSRC, "Academy of Medical Sciences / the Wellcome Trust / the Government Department of Business, Energy and Industrial Strategy / the British Heart Foundation / Diabetes UK" grants BSAC-2018–0095, NC/V001582/1, BB/V007823/1 and SBF006\1040, respectively (to RRMC), and the Swiss National Science Foundation Postdoc Mobility and Ambizione Fellowships P300PA_167703 and PZ00P3_180142, respectively (to DG).

## Additional information

### Funding

| Funder | Grant reference number | Author |
|---|---|---|
| Medical Research Council | MR/M009505/1 | Despoina AI Mavridou |
| Biotechnology and Biological Sciences Research Council | BB/M02623X/1 | Jessica MA Blair |
| Wellcome Trust | 105603/Z/14/Z | Gerald J Larrouy-Maumus |
| British Society for Antimicrobial Chemotherapy | BSAC-2018-0095 | Ronan R McCarthy |
| Biotechnology and Biological Sciences Research Council | BB/V007823/1 | Ronan R McCarthy |
| Swiss National Science Foundation | P300PA_167703 | Diego Gonzalez |
| National Centre for the Replacement, Refinement and Reduction of Animals in Research | NC/V001582/1 | Ronan R McCarthy |
| Biotechnology and Biological Sciences Research Council | BB/M011178/1 | Nikol Kaderabkova |
| Biotechnology and Biological Sciences Research Council | BB/M01116X/1 | Hannah L Pugh |
| Swiss National Science Foundation | PZ00P3_180142 | Diego Gonzalez |
| Academy of Medical Sciences | SBF006\1040 | Ronan R McCarthy |
| National Institutes of Health | R01AI158753 | Despoina AI Mavridou |

The funders had no role in study design, data collection and interpretation, or the decision to submit the work for publication.

## Author contributions

R Christopher D Furniss, Conceptualization, Formal analysis, Investigation, Methodology, Visualization, Writing – original draft, Writing – review and editing; Nikol Kaderabkova, Conceptualization, Formal analysis, Investigation, Methodology, Visualization, Writing – review and editing; Declan Barker, Evgenia Maslova, Amanda AA Antwi, Hannah L Pugh, Investigation, Writing – review and editing; Patricia Bernal, Methodology, Resources, Writing – review and editing; Helen E McNeil, Laurent Dortet, Resources, Writing – review and editing; Jessica MA Blair, Funding acquisition, Investigation, Methodology, Resources, Supervision, Writing – review and editing; Gerald Larrouy-Maumus, Formal analysis, Funding acquisition, Investigation, Writing – review and editing; Ronan R McCarthy, Funding acquisition, Investigation, Methodology, Supervision, Writing – review and editing; Diego Gonzalez, Formal analysis, Funding acquisition, Investigation, Methodology, Writing – review and editing; Despoina AI Mavridou, Conceptualization, Data curation, Formal analysis, Funding acquisition, Investigation, Methodology, Project administration, Supervision, Visualization, Writing – original draft, Writing – review and editing

## Author ORCIDs

R Christopher D Furniss http://orcid.org/0000-0002-5806-5099
Patricia Bernal http://orcid.org/0000-0002-6228-0496
Jessica MA Blair http://orcid.org/0000-0001-6904-4253
Despoina AI Mavridou http://orcid.org/0000-0002-7449-1151

## Decision letter and Author response

Decision letter https://doi.org/10.7554/eLife.57974.sa1
Author response https://doi.org/10.7554/eLife.57974.sa2

---

# Additional files

## Supplementary files

• Supplementary file 1. Analysis of the cysteine content and phylogeny of all identified β-lactamases. 6,649 unique β-lactamase protein sequences were clustered with a 90% identity threshold and the centroid of each cluster was used as a phylogenetic cluster identified for each sequence ("Phylogenetic cluster" column). All sequences were searched for the presence of cysteine residues ("Total number of cysteines" and "Positions of all cysteines" columns). Proteins with two or more cysteines after the first 30 amino acids of their primary sequence (cells shaded in grey in the "Number of cysteines after position 30" column) are potential substrates of the DSB system for organisms where oxidative protein folding is carried out by DsbA and provided that translocation of the β-lactamase outside the cytoplasm is performed by the Sec system. The first 30 amino acids of each sequence were excluded to avoid considering cysteines that are part of the signal sequence mediating the translocation of these enzymes outside the cytoplasm. Cells shaded in grey in the "Reported in pathogens" column mark β-lactamases that are found in pathogens or organisms capable of causing opportunistic infections. The Ambler class of each enzyme is indicated in the "Ambler class column" and each class (A, B1, B2, B3, C and D) is highlighted with a different color.

• Supplementary file 2. MIC data used to generate *Figure 1B*, *Figure 1—figure supplement 2*, and *Figure 5B*. Cells that are shaded in grey represent strain-antibiotic combinations that were not tested. The aminoglycoside antibiotic gentamicin serves as a control for all strains. For the "Supplementary File 2a" tab, values are representative of three biological experiments each conducted as a single technical repeat, and for the "Supplementary File 2b" tab, values are representative of two biological experiments each conducted as a single technical repeat.

• Supplementary file 3. Supplementary tables 1-6 and relevant citations.

• Transparent reporting form

## Data availability

All data generated during this study that support the findings are included in the manuscript or in the supplementary information.

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
