## [Editor Report]

This work is based on the idea that targeting protein stability or inhibiting proper protein folding in the membrane/periplasmic space might abolish antimicrobial resistances (AMR) in Gram-negative bacteria. By targeting the primary disulfide bond formation enzyme DsbA, the authors provide a proof-of-principle for the inhibition of β-lactamases, MCR enzymes, and RND efflux pumps of model bacterial species as well as clinical isolates. Collectively, the study shows that a chemical inhibitor of the DSB system sensitizes resistant bacteria to several antibiotics such as β-lactams, chloramphenicol, and colistin.

---

## [Decision Letter]

**Decision letter after peer review:**

Thank you for submitting your article "Breaking antimicrobial resistance by disrupting extracytoplasmic protein folding" for consideration by *eLife*. Your article has been reviewed by 3 peer reviewers, including Melanie Blokesch as Reviewing Editor and Reviewer #1, and the evaluation has been overseen by Wendy Garrett as the Senior Editor. The following individual involved in review of your submission has agreed to reveal their identity: Eric D Brown (Reviewer #3).

The reviewers have discussed the reviews with one another and the Reviewing Editor has drafted this decision to help you prepare a revised submission.

In this manuscript, the authors aim at identifying mechanism that counteract antimicrobial resistances (AMR) in Gram-negative bacteria. The study is based on the idea that targeting protein stability or inhibiting proper protein folding in the membrane/periplasmic space could abolish several AMRs all at once. For this reason, the authors targeted the primary disulfide bond formation enzyme DsbA. They provide proof-of-principle for the inhibition of β-lactamases, MCR enzymes, and RND efflux pumps using multiple species as well as clinical Gram-negative resistant isolates. Indeed, the study shows that a chemical inhibitor of the DSB system sensitizes resistant bacteria to several antibiotics such as β-lactams, chloramphenicol and colistin and that this combination treatment also works in a simplified in vivo infection model using the wax moth Galleria mellonella.

The experts considered the manuscript topical, interesting, and carefully done in general. However, they also identified several shortcomings that should be addressed before the study can be further considered for publication in *eLife*.

The main points are:

1) The presentation of the data needs to be improved. It is unclear why only representative experiments are shown and not individual values (potential on top of bars showing overall averages) including error bars. This is especially important, as some comparison show very small MIC differences and there is a substantial concern whether this reflects more noise than any significant difference, especially in case of the MCR part (the potential for β-lactamase inhibition seems much more solid). Indeed, we expect this to be corrected so that the reviewers can properly judge the significance of the differences and the real prospects for future drug development. As mentioned below in the minor points, the MICs are in part not "substantial" as claimed. This has to be addressed.

2) Statistics between dsbA alone and dsbA+pip data is needed for the Galleria panel in Figure 6. Indeed, it seems unlikely to be statistically significant. In this context, the reviewers considered it essential to include in vivo experiments for the combination of an anti-Dsb drug plus antibiotics to complement the in vivo data (as the in vivo part is also highlighted in the abstract).

3) Additional methods should be used to address the permeability possibility.

---

## [Author Response]

The main points are:1) The presentation of the data needs to be improved. It is unclear why only representative experiments are shown and not individual values (potential on top of bars showing overall averages) including error bars. This is especially important, as some comparison show very small MIC differences and there is a substantial concern whether this reflects more noise than any significant difference, especially in case of the MCR part (the potential for β-lactamase inhibition seems much more solid). Indeed, we expect this to be corrected so that the reviewers can properly judge the significance of the differences and the real prospects for future drug development. As mentioned below in the minor points, the MICs are in part not "substantial" as claimed. This has to be addressed.

We have included all experimental data in the revised manuscript. For minimum inhibitory concentration (MIC) assays, all recorded values are displayed in the relevant graphs. For assays where three or more biological experiments were performed, the bars indicate the median MIC value. For assays where two biological experiments were performed the bars indicate the most conservative of the two MIC values (i.e., for increasing trends, the value representing the smallest increase and for decreasing trends, the value representing the smallest decrease). We have avoided averaging our obtained MIC results because of the quantized nature of MIC assays, which only inform on bacterial survival for specific antibiotic concentrations and do not provide information for antibiotic concentrations that lie in-between the tested values. While this has precluded us from performing statistical tests for MIC assays (statistical analyses on MIC values are not common practice in the literature), it is clear from the presented graphs that even for modest effects (for example colistin MIC vales and MICs for effluxed antibiotics), our results are robust and reproducible. For all other assays, we have performed appropriate statistical analyses that are described in the “Materials and methods” section and further detailed in the figure legends.

Regarding the magnitude of the effects that we observe for different mechanisms of antibiotic resistance upon deletion of *dsbA*, please see our extensive reply to minor points 2 and 3, below.

2) Statistics between dsbA alone and dsbA+pip data is needed for the Galleria panel in Figure 6. Indeed, it seems unlikely to be statistically significant. In this context, the reviewers considered it essential to include in vivo experiments for the combination of an anti-Dsb drug plus antibiotics to complement the in vivo data (as the in vivo part is also highlighted in the abstract).

We apologise for this and thank the reviewers for their comment. Indeed, the *dsbA1* mutant of the *Pseudomonas aeruginosa* clinical isolate expressing OXA-198 that we used for the in vivo experiments in our submitted manuscript is not only affected in terms of antibiotic resistance, but also in terms of virulence. This is to be expected in *dsbA* mutants of *P. aeruginosa*, since DsbA is essential for the production of elastase (1, 2) that is crucial when this organism establishes an infection. As a result, when using this less virulent mutant to infect *Galleria mellonella*, it is not possible to achieve a comparable infection to that caused by the wild-type strain, something that then obscures our results during antibiotic treatment. To address the reviewers request, we first attempted to combine the available chemical inhibitor of the DSB system with appropriate antibiotics, in order to treat *G. mellonella* infected with resistant clinical strains of Enterobacteria used in Figure 6 of our study (the available chemical inhibitor is only appropriate for *E. coli* DsbB and its close analogues, and cannot inhibit DsbB proteins of *P. aeruginosa* (3)). While we obtained some promising trends from these experiments, they were not consistent enough for publication. This is due to the fact that the chemical compound that we use to inhibit the function of DsbA (the only available potent inhibitor of the DSB system) acts on DsbB. Inhibition of DsbB blocks the re-oxidation of DsbA and leads to its accumulation in its inactive reduced form. However, the action of the inhibitor can be bypassed through re-oxidation and re-activation of DsbA by small-molecule oxidants such as L-cystine (4), which are abundant in rich growth media or animal tissues. This makes the inhibitor only suitable for in vitro assays that can be performed in minimal media, where the presence of small-molecule oxidants can be strictly avoided, but entirely unsuitable for an insect or a vertebrate animal model.

Having observed that deletion of *dsbA1* does not equally affect the virulence of all clinical *P. aeruginosa* strains, we generated the *dsbA1* mutant of a different clinical isolate (*P. aeruginosa* PAe191 that produces OXA-19, the most disseminated OXA enzyme in clinical strains). Absence of DsbA in this strain caused a drastic reduction in the ceftazidime MIC value by over 220 µg/mL (Figure 7B of the revised manuscript), and the mutant strain was virulent enough to perform *G. mellonella* survival assays, allowing us to separate the effect of *dsbA1* deletion on antibiotic resistance from its effect on virulence. We found that deletion of *dsbA1* in combination with ceftazidime treatment led to increased survival of the larvae (17% mortality compared to 80% mortality for larvae infected with the *dsbA1* mutant but not treated with antibiotics, or 100% mortality at an earlier timepoint for larvae infected with the wild-type strain and treated with ceftazidime). Since survival assays are much more appropriate for the *G. mellonella* infection model and our results using this new set of strains were highly statistically significant, we have replaced our previous in vivo data with these newly obtained results (Figure 7C and lines 392-430 of the revised manuscript).

3) Additional methods should be used to address the permeability possibility.

1-N-phenylnaphthylamine (NPN) has been used extensively to assess outer membrane permeability in *E. coli*, both when testing mutant strains and when evaluating chemical permeabilizers. A few of the numerous examples found in the literature that give us confidence in using this assay, are given below:

a) Dose-dependent increase in NPN fluorescence was observed upon addition of increasing concentrations of colistin to *E. coli* strains. The authors concluded that NPN is a “quantitative read-out for colistin mediated outer membrane disruption” (5).

b) Dose-dependent increase in NPN fluorescence was observed as an increasing concentration of a novel outer membrane permeabilizing compound was added to *E. coli* strains. NPN uptake of outer-membrane-compromised mutants was also measured in the same study; namely a strain where CRISPRi was used to deplete *lolA* and a *waaG* mutant (6). In a different study, a strain expressing a version of LptD that lacks an extracellular loop key to its function, was also assessed using NPN (7), demonstrating that defects in LptD function would be clearly detectable in an NPN assay.

c) In a study examining the interaction of aminoglycoside antibiotics with the outer membrane of *E. coli* it was shown that gentamicin enhances NPN uptake in a concentration-dependent manner. Assays using lysozyme to tests outer membrane permeability provided comparable results with the performed NPN assays. The authors concluded that NPN is the “most sensitive” probe available for the assessment of outer membrane permeabilization (8).

To remove any doubts that the *dsbA* mutants used in our study do not have a cell envelope integrity defect, we performed additional assays on both *E. coli* K-12 *dsbA* mutants used in our manuscript. In particular, we complemented our NPN assays with vancomycin MIC assays to further test the integrity of the outer membrane. We also complemented our propidium iodide (PI) assays, which probe the integrity of the entire cell envelope, with assays based on the β-galactosidase substrate chlorophenyl red-β-D-galactopyranoside (CPRG), as requested by the reviewers. All four assays showed that the integrity of the outer membrane and of the cell envelope of the tested *dsbA* mutants is comparable to that of the parental strains (see Figure 1 - figure supplement 7 and Figure 1 - figure supplement 9 of the revised manuscript). These results are in agreement with the findings reported in (9), whereby only *surA* or *surAdsbA* mutants show outer membrane permeability defects due to impairment of LptD; the outer membrane integrity of the tested *dsbA* mutant is comparable to the parental strain. The authors argue that their results in this study are due to the fact that “the levels of LptD containing disulfides spontaneously formed by small oxidants present in the periplasm are sufficient”.